# $R^2$ec: Towards Large Recommender Models with Reasoning

**Runyang You**[1]    **Yongqi Li**[1*] **Xinyu Lin**[2]   **Xin Zhang**[1,4]
**Wenjie Wang**[3]   **Wenjie Li**[1]   **Liqiang Nie**[4]
[1]The Hong Kong Polytechnic University   [2]National University of Singapore
[3]University of Science and Technology of China   [4]Harbin Institute of Technology (Shenzhen)
runyang.y@outlook.com, liyongqi0@gmail.com,
xylin1028@gmail.com, izhx404@gmail.com,
wenjiewang96@gmail.com, cswjli@comp.polyu.edu.hk, nieliqiang@gmail.com

## Abstract

Large recommender models have extended LLMs as powerful recommenders via encoding or item generation, and recent breakthroughs in LLM reasoning synchronously motivate the exploration of reasoning in recommendation. In this work, we propose $R^2$ec, a unified large recommender model with intrinsic reasoning capability. $R^2$ec introduces a dual-head architecture that supports both reasoning chain generation and efficient item prediction in a single model, significantly reducing inference latency. To overcome the lack of annotated reasoning data, we design RecPO, a reinforcement learning framework that optimizes reasoning and recommendation jointly with a novel fused reward mechanism. Extensive experiments on three datasets demonstrate that $R^2$ec outperforms traditional, LLM-based, and reasoning-augmented recommender baselines, while further analyses validate its competitive efficiency among conventional LLM-based recommender baselines and strong adaptability to diverse recommendation scenarios. Code and checkpoints available at https://github.com/YRYangang/RRec.

## 1   Introduction

Large language models (LLMs) have demonstrated remarkable capabilities in contextual understanding and open-ended generation [1–5]. This success has catalyzed the development of large recommender models that inherit and specialize in the advantages of LLMs for recommendation. Current approaches can be divided into two main streams: one employs LLMs as powerful encoders to embed users (their historical interactions) [6–8], while the other reformulates item prediction into the autoregressive generation of item identifiers [4, 9]. These large recommender models exhibit remarkable generalization capabilities, achieving advanced performance across diverse application scenarios, including cold-start recommendations [10], cross-domain personalization [11], and long-tail item prediction [12, 13].    The frontier of LLM capabilities now extends beyond model size scaling to test-time scaling, *i.e.,* reasoning. Emerging advances like DeepSeek-R1 [14] demonstrates that such extra computation during inference can further improve LLM capabilities in areas such as mathematics, coding, and scientific problem-solving [15]. Given that large recommender models are instantiated from pretrained LLMs, a natural question is: how can large recommender model benefit from reasoning to further enhance recommendation performance?

Existing studies have preliminarily explored LLM reasoning for recommendation, including user preference analysis [16, 17], synthetic item profile enrichment [18, 19], user search query rewriting [20, 21], and rating prediction [22]. These approaches typically take LLMs as additional reason-

---

*Yongqi Li is the corresponding author.

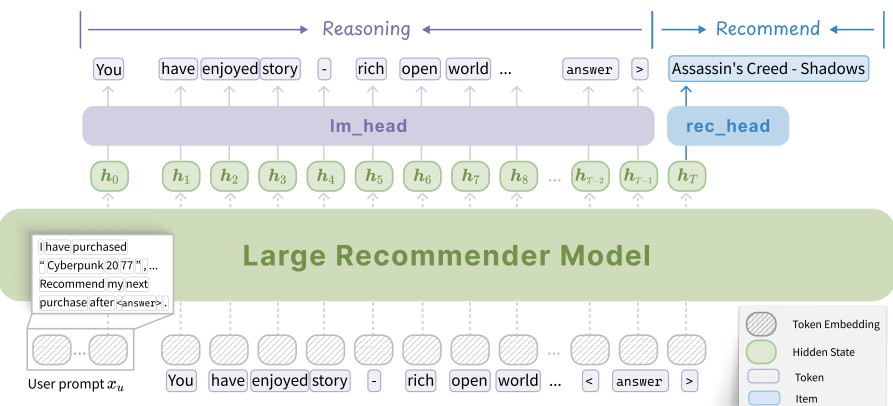

Figure 1: The architecture overview of R²ec, which facilitates interleaved reasoning and recommendation in an autoregressive process with two task-specific heads: 1) language-modeling head (`lm_head`) for reasoning generation; and 2) recommendation head (`rec_head`) for item prediction.

ing modules to augment the original recommendation pipeline, decoupling reasoning from item prediction. However, such designs introduce crucial limitations: 1) Significant resource cost. Two distinct modules, namely the large reasoning model and the recommendation model, must be trained, checkpointed, and served in parallel, inflating both memory footprint and inference latency. 2) Suboptimal joint optimization. Reasoning and recommendation modules can only be trained by freezing one while updating the other [16, 23]. This alternate optimization scheme prevents gradient flow across the pipeline, precluding true end-to-end learning, hindering fine-grained alignment between reasoning and ranking objectives, and ultimately leading to suboptimal convergence. In this work, we aim to develop a unified large recommender model with intrinsic reasoning capabilities, exploring new technical solutions for reasoning-enhanced recommendation systems.

It is non-trivial to develop a unified large recommender model with reasoning due to the following aspects: 1) Model design. Most large recommender models rely on autoregressive decoding of item identifiers (IDs), which can be computationally expensive [1–4, 24–26]. Introducing reasoning would inevitably exacerbate the latency. Thus, a key challenge is to integrate reasoning while maintaining acceptable inference speed. 2) Training optimization. A major obstacle in training the reasoning recommender is the scarcity of high-quality, annotated reasonings in recommendation domains. Unlike general QA tasks where solution rationales can be curated, the underlying rationale for recommendation is inherently subjective and difficult to collect at scale. Reinforcement Learning (RL) [21, 27, 28] is a natural alternative, but it introduces recommendation-specific challenges in reward specification and objective design. The training framework must effectively couple the reasoning and recommendation objectives to ensure co-optimization.

To address the above issues, we propose a unified large **Rec**ommender model with intrinsic **R**easoning, dubbed as **R²ec**. Our approach is built on two pillars: 1) Model design. As illustrated in Figure 1, we equip an LLM with a dual-head design: a standard language head for reasoning and a novel recommendation head for item prediction. The recommendation head maps the model's output to a shared semantic item embedding space. This allows the model to first generate a reasoning chain autoregressively and then select the final item in a single step, replacing the slow auto-regressive decoding of item IDs with a more efficient "next-item prediction". 2) Optimization. we propose **RecPO**, an RL-based training framework without human reasoning annotations. First, for a given input, we sample multiple diverse reasoning paths, each culminating in an item recommendation, forming "reasoning-then-item" sequences analogous to those in reasoning LLMs. Second, we found it insufficient to set the reward naively as ranking metrics, we thus propose a fused reward scheme that combines discrete ranking rewards with continuous similarity rewards. Finally, we formulate a joint training objective that leverages the rewards to optimize the model, encouraging it to develop reasoning pathways that lead to high-quality recommendations.

We conducted extensive experiments to evaluate the efficacy of R²ec by comparing it with various traditional, LLM-based, and reasoning-augmented recommendation baselines. The experimental results on three datasets demonstrate that R²ec significantly outperforms all baseline methods, verifying the effectiveness of its unified architect and its ability to leverage reasoning for superior recommendation quality. Our comprehensive evaluation comprises **nine** analyses, including a detailed

profiling of the model's reasoning behavior which reveals its context-aware capabilities; an efficiency analysis showing competitive inference speed among LLM-based recommenders; ablation and optimization studies on key components; and additional investigations into scalability, generalization and case studies.

Key contributions are as follows:

- We introduce $R^2$ec, the first unified large recommender model that intrinsically integrates reasoning and recommendation within a single architecture, resulting in significantly lower inference latency compared to both reasoning-augmented and conventional LLM-based recommender baselines.
- We propose RecPO, an RL framework for recommendation that enables model optimization without reasoning annotations. We derive a joint optimization objective and propose a fused reward scheme, yielding performance that significantly surpasses all baselines.
- Beyond metrics, we provide an in-depth analysis of the model's reasoning behavior, demonstrating that $R^2$ec automatically develops context-aware reasoning strategies tailored to specific domains and user contexts, offering meaningful interpretability for its recommendations.

## 2 Preliminaries

### 2.1 LLM-based Recommendation

LLM-based recommendation is an emerging paradigm, and the most applicable approach is to leverage LLMs as encoders to embed users and items, which are widely adopted in industry [29, 24, 30]. Typically, the recommendation process within this paradigm involves the following structured steps:

**User and Item Formulation.** Structured interaction histories and item metadata are first formulated into natural language prompts suitable for LLMs. We first collect a user's historically interacted items with the corresponding ratings. We then construct a textual prompt includes both the instruction and a natural-language description of these past behaviors. *E.g., " User has purchased and rated the following products sequentially: 1. 'Avatar Blu-ray' (4/5); 2. 'Wireless Headphones' (5/5); . . . Recommend the next item."* Finally, we tokenize this prompt into a sequence of tokens, denoted as $x_u$, which serves as the LLM input. Likewise, each candidate item $v$ can be described using its metadata, then tokenized into a sequence of tokens $x_v$.

The tokenized textual prompts $x_u$ and $x_v$ are then input into the LLM transformer backbone, by extracting the final hidden state of the input sequence, we can obtain corresponding user embedding $\boldsymbol{h}_u$ and item embedding $\boldsymbol{h}_v$.

**Training via Contrastive Learning.** To optimize LLM for recommendation tasks, in-batch contrastive loss is typically adopted:

$$\mathcal{L}_{\text{CL}} = -\log \frac{\exp(\boldsymbol{h}_u^\top \boldsymbol{h}_{v^+}/\tau)}{\sum_{v' \in B} \exp(\boldsymbol{h}_u^\top \boldsymbol{h}_{v'}/\tau)}, \tag{1}$$

where $\tau$ is the temperature hyperparameter controlling similarity dispersion, $B$ is the ground truth item set in one batch, and $v^+$ is the groud-truth target item.

**Inference.** During inference, recommendation score $s(u, v)$ for each user-item pair is computed via the inner product between their embeddings: $s(u, v) = \boldsymbol{h}_u^\top \boldsymbol{h}_v$. Finally, we rank all items according to the scores and obtain top-$K$ item recommendation list.

### 2.2 Reinforcement Learning for Large Reasoning Models

**Terminologies and Notations.** RL for LLMs treats text generation as a sequential action process, where each action refers to outputting the next token. Formally, at each time step $t$, we denote the current state as $(x, o_{<t})$, where $x$ is the initial prompt and $o_{<t}$ is all previously generated tokens. Then, the conditioned probability distribution of an action is denoted as $\pi_\theta(o_t|x, o_{<t})$, where $\pi_\theta$ is the policy parameterized by $\theta$. Based on the distribution, we can obtain the next token $o_t$ through manual selection, greedy or temperature sampling. Upon obtaining a complete sequence $o$, termed as a trajectory, a reward function then evaluates and assigns it with a scalar score $R$ that can reflect

user satisfaction or answer correctness. However, using $R$ directly in gradient estimates leads to high variance and unstable updates; instead, the advantage $A$, which measures how much better an action is than the expected return from that state, is adopted to reduce variance and improve sample efficiency [27, 28, 31, 32].

**Sampling and Advantage Computation.** Sampling-based advantage estimation is widely adopted in recent advances [27, 31–33]; below we describe its basic pipeline and two typical computation methods. Given an input $x$, a group of $G$ different trajectories $\{o_i\}_{i=1}^G$, are sampled from $\pi_{\theta_{old}}$. Existing studies widely obtain these trajectories via top-$K$ sampling with temperature, where $\theta_{old}$ refers to the frozen policy parameters before current update. Each trajectory $o_i$ receives a scalar reward $R_i$, which will then be used to compute the trajectory-level advantages via two widely adopted approaches, namely GRPO [27] and RLOO [31]:

$$A_i^{\text{RLOO}} = R_i - \frac{1}{G-1}\sum_{j \neq i}^{G} R_j, \quad A_i^{\text{GRPO}} = \frac{R_i - \text{mean}(\{R_j\}_{j=1}^G)}{\text{std}(\{R_j\}_{j=1}^G)}. \tag{2}$$

**Training Objective.** Training then proceeds with policy-gradient algorithms. Specifically, let

$$r_{i,t}(\theta) = \frac{\pi_\theta(o_{i,t}|x, o_{i,<t})}{\pi_{\theta_{old}}(o_{i,t}|x, o_{i,<t})} \tag{3}$$

be the importance ratio between the updated and old policies of trajectory $i$ at token position $t$. The training objective is given by:

$$\mathcal{J}(\theta) = \mathbb{E}_{x \sim \mathcal{D}, \{o_i\}_{i=1}^G \sim \pi_{\theta_{old}}(\cdot|x)} \sum_{i=1}^{G} \sum_{t=1}^{|o_i|} [\min(r_{i,t}(\theta)A_i, \text{clip}(r_{i,t}(\theta), 1-\epsilon, 1+\epsilon)A_i)], \tag{4}$$

where $\epsilon$ defines the clipping trust-region radius, which prevents excessively large updates, thereby reducing variance and improving optimization stability.

# 3  $R^2$ec: Large Recommender Model with Reasoning

$R^2$ec is a large recommender model that "thinks" to recommend. We first outline the model design that enables the incorporation of generative reasoning and discriminative recommendation into a single model in Section 3.1. And then our proposed RL optimization algorithm tailored for achieving unified reasoning and recommendation capabilities is introduced in Section 3.2.

## 3.1  Model Design

**Architecture.** As illustrated in Figure 1, our proposed $R^2$ec is built upon a decoder-only backbone with two task-specific heads: 1) *Language-modeling head* (`lm_head`). The token embedding table $\mathbf{H}_\mathcal{T} \in \mathbb{R}^{|\mathcal{T}| \times d}$. Where $\mathcal{T}$ is the token set, and each row is a $d$-dimensional embedding for one token. This head is responsible for the generation of reasoning tokens. 2) *Recommendation head* (`rec_head`). The item embedding table $\mathbf{H}_\mathcal{V} \in \mathbb{R}^{|\mathcal{V}| \times d}$. Where $\mathcal{V}$ is the item set, and each row $\boldsymbol{h}_v$ in $\mathbf{H}_\mathcal{V}$ is obtained by encoding item description prompt into the model itself and extracting the final hidden state. This head is used to score items for recommendation. Such design integrates generative reasoning and discriminative recommendation within a single unified model. It supports flexible and scalable item update by simply adding, deleting, or replacing vectors in the item embedding table, contrasting with generative recommendation systems that require hard-coded tokenization [34, 9, 4, 35], enabling effective zero-shot generalization and accommodating large-scale item catalogs without severe degradation on recommendation quality or efficiency.

**Item prediction.** Inference begins with feeding the tokenized user prompt $x_u$ (template of prompts can be found in Appendix G) into the transformer-based backbone, producing an initial hidden state $\boldsymbol{h}_0$. The language-modeling head then maps $\boldsymbol{h}_0$ to the first reasoning token $o_1$. This process continues autoregressively, yielding a sequence of $T$ reasoning tokens $o_{1:T}$. The final hidden state of the generated sequence $\boldsymbol{h}_T$ is then fed into the recommendation head, where each candidate

item $v \in \mathcal{V}$ is scored by an inner product $s(v) = \boldsymbol{h}_T^\top \mathbf{H}_\mathcal{V}[v]$, $v \in \mathcal{V}$, which determines the final ranking. This mechanism yields a tight reasoning–recommendation coupling, since both the language-modeling head and recommendation head share the same hidden-state space as input, reasoning directly reshapes $\boldsymbol{h}_T$ and thus yielding more accurate recommendation scores $s(v)$. Such alignment ensures that reasoning optimization (Section 3.2) contributes directly to finer recommendation.

## 3.2 Optimization: RecPO

The goal is to train the policy $\pi_\theta$ to jointly perform reasoning and recommendation, *i.e.,* it must generate coherent reasoning sequences to rank the target item accurately. Accordingly, we structure our optimization workflow in three parts. 1) we introduce the *trajectory sampling* strategy that draws multiple reasoning trajectories for each user (Section 3.2.1). 2) we describe *reward and advantage estimation*, where discrete ranking signals and softmax similarities are fused into a single scalar award for the above sampled trajectories (Section 3.2.2). 3) we formulate the *training objective*, which blends reasoning-level and recommendation-level updates through a clipped-ratio loss (Section 3.2.3). Complete description of training and inference pipeline are in Appendix E.

### 3.2.1 Trajectory Sampling

Due to the optimization objective of joint learning, we define one trajectory in our settings spanning the entire *reasoning-then-recommend* process:

$$x_u \xrightarrow{\pi_\theta} o_1 \xrightarrow{\pi_\theta} \dots \xrightarrow{\pi_\theta} o_T \xrightarrow{\pi_\theta} v^+,$$

where the initial state $x_u$ encodes the user history and instruction, $o_1, ..., o_T$ represents the intermediate actions of outputting $T$ reasoning tokens, and $v^+$ as the action of recommending the ground-truth target item.

For each user $u$, we first sample $G$ reasonings $\{o_i\}_{i=1}^G$ with the old policy $\pi_{\theta_{\text{old}}}$ using the tokenized input $x_u$: $\{o_i\}_{i=1}^G \sim \pi_{\theta_{\text{old}}}(\cdot \mid x_u)$ by top-$K$ sampling with temperature to control stochasticity. Each sampled reasoning is then fed through the policy $\pi_\theta$ to produce a complete reasoning-then-recommend trace, which is subsequently used for reward calculation and advantage estimation.

### 3.2.2 Reward and Advantage Estimation

Given the above sampled trajectories, we now aim to assign rewards to them. Basically, the reward should align with the evaluation criteria, *i.e.,* the recommendation metrics in our work, encouraging the model to achieve better performance. However, in practice, we find that directly using the recommendation metrics as rewards is insufficient, as many trajectories of varying quality can result in the same top-$K$ ranking. We therefore introduce a fused reward scheme that combines discrete ranking reward $R_d$ and continuous similarity reward $R_c$, which are formulated as follows:

$$R_d = \text{NDCG} @k \left( \text{rank}(v^+) \right), \qquad R_c = \frac{\exp\left(\boldsymbol{h}_T^\top \boldsymbol{h}_{v^+}/\tau\right)}{\sum_{v \in \mathcal{V}} \exp\left(\boldsymbol{h}_T^\top \boldsymbol{h}_v/\tau\right)},$$

where $R_d$ is the NDGC, $R_c$ is the softmax similarity of recommending target item against all items in $\mathcal{V}$. The final reward is then obtained through a linear combination:

$$R = \beta R_c + (1 - \beta) R_d, \qquad \beta \in [0, 1], \tag{5}$$

where the weighting coefficient $\beta$ is empirically set to $\beta \approx 0.05$ to keep the ranking term dominant while providing sufficient resolution among trajectories that attain identical ranks. With rewards $\{R_i\}_{i=1}^G$ we can obtain trajectory-level advantages $\{A_i\}_{i=1}^G$ via Eqn.(2).

### 3.2.3 Training Objective

Given the goal of joint optimization of reasoning and recommendation, we treat the *entire* reasoning-then-recommend sequence $(x_u, o_1, ..., o_T, v^+)$ as a single RL trajectory. Policy optimization therefore operates over a composite action space, where the policy first makes token-level decisions to generate reasoning, then selects an item at the recommendation stage. Under this formulation, the importance ratio from Eqn. (3) extends to:

$$r_{i,t}(\theta) = \begin{cases} \frac{\pi_\theta(o_{i,t}|x_u, o_{i,<t})}{\pi_{\theta_{\text{old}}}(o_{i,t}|x_u, o_{i,<t})}, & \text{if } t \leqslant T \text{ (reasoning)} \\ \frac{\pi_\theta(v^+|x_u, o_{i,\leqslant T})}{\pi_{\theta_{\text{old}}}(v^+|x_u, o_{i,\leqslant T})}, & \text{if } t = T + 1 \text{ (recommendation)}. \end{cases} \tag{6}$$

Specifically, we model the recommending action, *i.e.,* recommending the target item $v^+$ via the in-batch softmax:

$$\pi_\theta(v^+ \mid x_u, o_i) = \frac{\exp(\boldsymbol{h}_T^\top \boldsymbol{h}_{v^+}/\tau)}{\sum_{v' \in B} \exp(\boldsymbol{h}_T^\top \boldsymbol{h}_{v'}/\tau)}. \tag{7}$$

Let $\ell_\epsilon(r, A) = \min(rA, \ \text{clip}(r, 1 - \epsilon, 1 + \epsilon)A)$ be the standard clipping operator with threshold $\epsilon$, we define the joint reasoning-and-recommendation training objective as:

$$\mathcal{J}(\theta) = \mathbb{E}_{\{u,v^+\}\sim\mathcal{D},\{o_i\}_{i=1}^G\sim\pi_{\theta_{\text{old}}}(\cdot|x_u)} \frac{1}{G} \sum_{i=1}^G \Big[\sum_{t=1}^{T_i} \ell_\epsilon(r_{i,t}(\theta), A_i) + \delta_{i,i^\star}\ell_\epsilon\big(r_{i,T+1}(\theta), A_i\big)\Big]. \tag{8}$$

In Eqn.(8), all trajectories contribute to token-level policy updates via the standard clipped objective $\ell_\epsilon\big(r_{i,t}(\theta), A_i\big)$. This design ensures the policy continues to learn from diverse reasoning behaviors. For the last recommendation action, only the trajectory with the highest advantage, identified by $i^\star = \arg\max_j A_j$, contributes gradients to recommendation optimization. This design concentrates recommendation learning with the most promising reasoning path, while the presence of other $G - 1$ reasoning paths contributes to ample exploration in reasoning actions, thus preserving exploration and ensuring effective recommendation learning. We further provide a gradient-based explanation of how item encoding, as a core component of the model's recommendation ability, participates in the objective and enables end-to-end optimization in Appendix F.

# 4 Experiments

## 4.1 Setups

**Dataset and Metrics.** Following previous works [3, 2, 1, 25], we conducted experiments using real-world datasets sourced from Amazon, including CDs and Vinyl (CDs), Video Games (Games), and Musical Instruments (Instruments). Dataset statistics and preprocessing steps detailed in Appendix B. We utilized two commonly used metrics: *Hit Rate* (H@K) and *Normalized Discounted Cumulative Gain* (N@K), with cutoff $K$ set to 5, 10 and 20. We adopted the full set recommendation setting, where metrics are computed over the entire item set, providing a better reflection of practical scenarios.

**Baselines and Implementation.** We selected competitive baselines from various categories, including traditional sequential recommenders (GRU4Rec [36], Caser [37], SASRec [38]), LLM-based recommender (TIGER [9], BigRec [2], $D^3$ [3], SDPO [25], Llara [1], SPRec [39]), and reasoning augmented recommendation systems (LangPTune [16]). For fair comparison, we adapted LLaRA and SDPO by removing candidate prompts to perform constrained generation over the full item corpus, denoted as LLaRA* and SDPO* respectively. Backboned on `Gemma2-2B-Instruct` and `Qwen2.5-3B-Instruct`. More baseline and implementation details can be found in Appendix C and D, respectively.

## 4.2 Overall Performance

To validate the effectiveness of the proposed $R^2$ec, we showed the overall performance of baselines and our $R^2$ec in Table 1. By analyzing the results, we gained the following findings. 1) Overall, $R^2$ec consistently outperforms every competing baseline, underscoring the value of jointly optimizing reasoning and recommendation. 2) Traditional methods perform well on the Instruments dataset but struggle on CDs and Games, revealing their limited generality. LangPTune frequently ranks second, which validates the benefit of integrating explicit reasoning into the recommendation pipeline. Finally, large recommenders generally outperform traditional approaches, notably secure secondary position on the Games dataset. We attribute this advantage to their larger model scale and semantic understanding capabilities. 3) Comparing two backbones, Gemma consistently outperforms its larger counterpart, achieving up to 2× gains for $D^3$ – suggesting that Gemma may generally deliver stronger recommendation performance despite its smaller parameter count (2B vs. 3B).

## 4.3 Ablation Study

We conducted ablation studies by evaluating the following variants: 1) "*w/ ClsHead*", an alternative design that uses classification head to the hidden state of reasoning tokens, instead of using the unified

Table 1: The overall performance of baselines and R$^2$ec on three datasets. The best results in each group are marked in Bold, while the second-best results are underlined. * implies the improvements over the second-best results are statistically significant (p-value < 0.05). % improve represents the relative improvement achieved by R$^2$ec over the best baseline.

| | | Instruments | | | | | | CDs and Vinyl | | | | | | Video Games | | | | | |
|---|---|---|---|---|---|---|---|---|---|---|---|---|---|---|---|---|---|---|---|
| **Method** | | H@5 | N@5 | H@10 | N@10 | H@20 | N@20 | H@5 | N@5 | H@10 | N@10 | H@20 | N@20 | H@5 | N@5 | H@10 | N@10 | H@20 | N@20 |
| GRU4Rec | | 0.0171 | 0.0135 | 0.0193 | 0.0142 | 0.0201 | 0.0144 | 0.0067 | 0.0037 | 0.0104 | 0.0041 | 0.0156 | 0.0051 | 0.0109 | 0.0070 | 0.0181 | 0.0093 | 0.0301 | 0.0123 |
| Caser | | 0.0109 | 0.0141 | 0.0115 | 0.0149 | 0.0127 | 0.0155 | 0.0045 | 0.0029 | 0.0067 | 0.0037 | 0.0089 | 0.0042 | 0.0124 | 0.0083 | 0.0191 | 0.0103 | 0.0279 | 0.0126 |
| SASRec | | 0.0175 | 0.0144 | 0.0201 | 0.0162 | 0.0223 | 0.0210 | 0.0076 | 0.0104 | 0.0081 | 0.0119 | 0.0086 | 0.0141 | 0.0129 | 0.0080 | 0.0206 | 0.0105 | 0.0326 | 0.0135 |
| TIGER | | 0.0171 | 0.0128 | 0.0184 | 0.0132 | 0.0193 | 0.0134 | 0.0067 | 0.0045 | 0.0097 | 0.0055 | 0.0156 | 0.0069 | 0.0123 | 0.0085 | 0.0222 | 0.0116 | 0.0323 | 0.0142 |
| BigRec | *Qwen* | 0.0052 | 0.0033 | 0.0111 | 0.0052 | 0.0189 | 0.0072 | 0.0045 | 0.0025 | 0.0089 | 0.0039 | 0.0141 | 0.0052 | 0.0008 | 0.0004 | 0.0016 | 0.0006 | 0.0128 | 0.0034 |
| $D^3$ | | 0.0042 | 0.0020 | 0.0094 | 0.0037 | 0.0192 | 0.0062 | 0.0082 | 0.0057 | 0.0141 | 0.0076 | 0.0253 | 0.0104 | 0.0054 | 0.0028 | 0.0104 | 0.0044 | 0.0197 | 0.0067 |
| LangPTune | | 0.0127 | 0.0083 | 0.0224 | 0.0115 | 0.0348 | 0.0145 | 0.0074 | 0.0053 | 0.0156 | 0.0080 | 0.0208 | 0.0094 | 0.0049 | 0.0027 | 0.0088 | 0.0040 | 0.0140 | 0.0140 |
| **R$^2$ec** | | **0.0237\*** | **0.0154\*** | **0.0374\*** | **0.0198\*** | **0.0615\*** | **0.0259\*** | **0.0513\*** | **0.0372\*** | **0.0647\*** | **0.0414\*** | **0.0818\*** | **0.0457\*** | **0.0288\*** | **0.0185\*** | **0.0532\*** | **0.0264\*** | **0.0827\*** | **0.0337\*** |
| *% Improve.* | | 35.43% | 6.94% | 66.96% | 22.22% | 52.61% | 23.33% | 46.57% | 58.30% | 37.95% | 51.09% | 20.83% | 40.62% | 42.36% | 34.05% | 51.13% | 41.29% | 31.56% | 33.53% |
| BigRec | *Gemma* | 0.0068 | 0.0048 | 0.0101 | 0.0058 | 0.0130 | 0.0066 | 0.0030 | 0.0030 | 0.0052 | 0.0037 | 0.0119 | 0.0053 | 0.0156 | 0.0105 | 0.0260 | 0.0138 | 0.0430 | 0.0182 |
| $D^3$ | | 0.0072 | 0.0038 | 0.0202 | 0.0080 | 0.0339 | 0.0114 | 0.0216 | 0.0129 | 0.0327 | 0.0164 | 0.0446 | 0.0194 | 0.0117 | 0.0068 | 0.0210 | 0.0141 | 0.0478 | 0.0224 |
| SDPO* | | 0.0066 | 0.0034 | 0.0098 | 0.0054 | 0.0144 | 0.0071 | 0.0022 | 0.0018 | 0.0037 | 0.0025 | 0.0162 | 0.0094 | 0.0166 | 0.0122 | 0.0298 | 0.0155 | 0.0466 | 0.0222 |
| Llara* | | 0.0078 | 0.0055 | 0.0137 | 0.0074 | 0.0159 | 0.0080 | 0.0097 | 0.0039 | 0.0127 | 0.0049 | 0.0202 | 0.0152 | 0.0275 | 0.0173 | 0.0428 | 0.0223 | 0.0677 | 0.0299 |
| SPRec | | 0.0070 | 0.0033 | 0.0111 | 0.0062 | 0.0142 | 0.0077 | 0.0029 | 0.0022 | 0.0037 | 0.0025 | 0.0124 | 0.0063 | 0.0152 | 0.0113 | 0.0244 | 0.0133 | 0.0566 | 0.0211 |
| LangPTune | | 0.0130 | 0.0079 | 0.0221 | 0.0107 | 0.0403 | 0.0152 | 0.0350 | 0.0235 | 0.0469 | 0.0274 | 0.0677 | 0.0325 | 0.0068 | 0.0053 | 0.0120 | 0.0059 | 0.0195 | 0.0094 |
| **R$^2$ec** | | **0.0264\*** | **0.0161\*** | **0.0397\*** | **0.0203\*** | **0.0615\*** | **0.0257\*** | **0.0573\*** | **0.0398\*** | **0.0804\*** | **0.0472\*** | **0.1042\*** | **0.0527\*** | **0.0326\*** | **0.0205\*** | **0.0531\*** | **0.0271\*** | **0.0835\*** | **0.0347\*** |
| *% Improve.* | | 50.86% | 11.81% | 77.23% | 25.31% | 52.61% | 22.38% | 63.71% | 69.36% | 71.43% | 72.26% | 53.91% | 62.15% | 18.98% | 19.19% | 24.07% | 21.52% | 23.34% | 16.25% |

Table 2: Ablation study on key components of R$^2$ec.

| | Instruments | | | | | | CDs and Vinyl | | | | | | Video Games | | | | | |
|---|---|---|---|---|---|---|---|---|---|---|---|---|---|---|---|---|---|---|
| **Method** | H@5 | N@5 | H@10 | N@10 | H@20 | N@20 | H@5 | N@5 | H@10 | N@10 | H@20 | N@20 | H@5 | N@5 | H@10 | N@10 | H@20 | N@20 |
| *w/ ClsHead* | 0.0044 | 0.0023 | 0.0102 | 0.0033 | 0.0179 | 0.0067 | 0.0030 | 0.0025 | 0.0045 | 0.0027 | 0.0095 | 0.0044 | 0.0012 | 0.0008 | 0.0022 | 0.0011 | 0.0133 | 0.0032 |
| *w/o Reasoning* | 0.0176 | 0.0121 | 0.0296 | 0.0153 | 0.0511 | 0.0200 | 0.0469 | 0.0321 | 0.0692 | 0.0393 | 0.0945 | 0.0456 | 0.0277 | 0.0174 | 0.0441 | 0.0227 | 0.0748 | 0.0303 |
| *w/o $R_d$* | 0.0198 | 0.0124 | 0.0338 | 0.0164 | 0.0560 | 0.0224 | 0.0521 | 0.0338 | 0.0766 | 0.0404 | 0.0974 | 0.0486 | 0.0302 | 0.0196 | 0.0487 | 0.0254 | 0.0798 | 0.0332 |
| *w/o $R_c$* | 0.0244 | 0.0160 | 0.0394 | **0.0208** | 0.0605 | **0.0258** | 0.0543 | 0.0382 | 0.0774 | 0.0456 | 0.1012 | 0.0515 | 0.0316 | 0.0202 | **0.0534** | 0.0264 | 0.0814 | 0.0355 |
| **R$^2$ec** | **0.0264** | **0.0161** | **0.0397** | 0.0203 | **0.0615** | 0.0257 | **0.0588** | **0.0388** | **0.0804** | **0.0457** | **0.1086** | **0.0525** | **0.0326** | **0.0205** | 0.0531 | **0.0271** | **0.0853** | **0.0363** |

reasoning–recommendation formulation. 2) "*w/o Reasoning*", where reasoning tokens are removed from prompts and the model is trained solely with in-batch contrastive loss; 3) "*w/o $R_c$*", which retains only the discrete ranking reward $R_d$ while removing the continuous similarity reward $R_c$ in Eqn. (5); 4) "*w/o $R_d$*", which removes the discrete ranking reward $R_d$ from Eqn. (5); and The results are summarized in Table 2, and several observations stand out.

1) It is found that R$^2$ec achieves an average improvement of roughly 15% across all metrics compared to *w/o Reasoning*. These substantial gains demonstrate that our designed optimization have enabled R$^2$ec to better leverage test-time scaling to deliver significantly stronger recommendation performance. 2) The *w/ ClsHead* variant substantially underperforms our tightly-coupled architecture, highlighting the necessity of reasoning–recommendation co-optimization. 3) As revealed that *w/o $R_c$* (using only $R_d$) consistently outperforms *w/o $R_d$* (using only $R_c$), this indicates that adopting reward signal that directly reflects evaluation result is crucial for training, while the continuous reward $R_c$, despite offering finer granularity, fails to provide meaningful distinctions and instead introduces noise that leads to suboptimal performance. 4) By fusing $R_d$ with a small weight on $R_c$, our approach preserves the task alignment of ranking rewards while benefiting from the supplementary signal of the continuous term. As a result, R$^2$ec achieves optimal performance on nearly all metrics.

## 4.4 Analysis on Optimization

This subsection presents analyses of key optimization strategies in RecPO, evaluating their impact on final performance.

### 4.4.1 Analysis on Advantage Estimation

Accurate advantage estimation is crucial for reducing variance and improving sample efficiency in policy-gradient RL [31, 28, 32]. We therefore evaluated two estimators, GRPO [40] and RLOO [31],

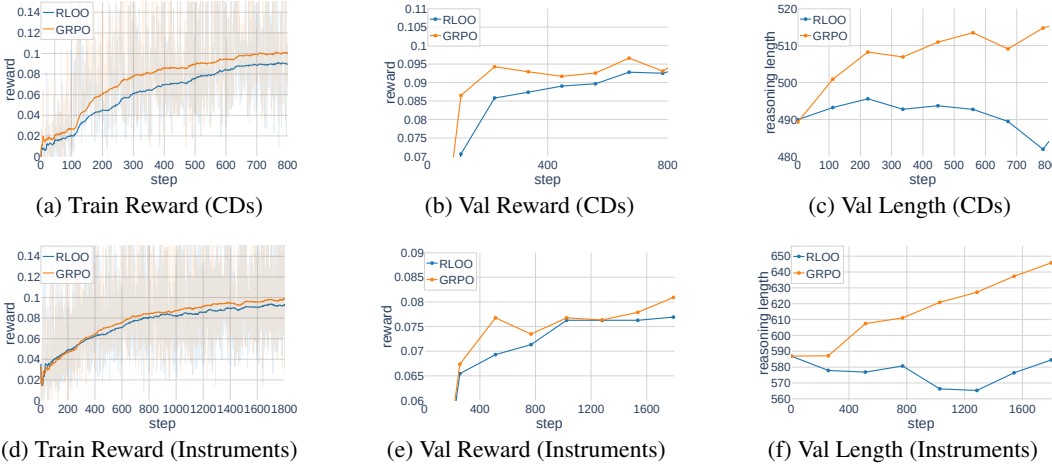

Figure 2: Analysis on advantage estimation methods, RLOO and GRPO, across two datasets. "Train Reward" and "Val Reward" indicate the variation in rewards on the training set and validation set, respectively. "Val Length" represents the variation in reasoning length on the validation set.

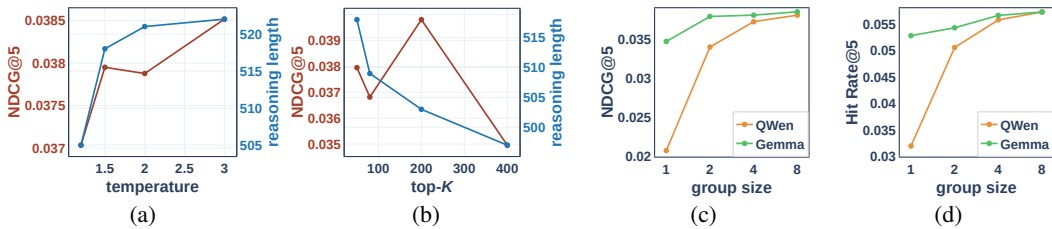

Figure 3: Analysis on trajectory sampling and group size over the CDs dataset. (a) and (b) show the impact of temperature and top-$K$ sampling on performance and reasoning length, respectively. (c) and (d) present the effect of group size on NDCG@5 and Hit Rate@5, respectively.

in our training pipeline. The variations of the training reward, validation reward, and reasoning length across the training steps are summarized in Figure 2, and we had the following observations.

1) First, as shown in Figure 2a and Figure 2d, both RLOO and GRPO exhibit high-variance training curves. This is due to the inherent nature of recommendation environments, which produce highly varied reward magnitudes—some sessions result in high rankings, while others yield very low ones. 2) Second, as Figure 2b and Figure 2e illustrate, GRPO demonstrates faster learning in the initial training steps and consistently outperforms RLOO in terms of validation reward, whereas RLOO progresses more steadily. This divergence stems from the GRPO's unit-variance normalization magnifies rewards into larger gradients that accelerate early learning. 3) Besides, as shown in Figure 2c and Figure 2f, GRPO's reasoning length gradually increases as training progresses, which is consistent with the phenomenon observed in LLM reasoning training [40], while RLOO appears to maintain a certain level of stability. This is because RLOO-driven optimization does not encourage improved reasoning with the low reward magnitudes in our task.

#### 4.4.2 Analysis on Trajectory Sampling

During RecPO, we performed trajectory sampling via the temperature $\tau$ and top-$K$ sampling, which influence the stochasticity and diversity of the generated samples. To quantify their impact, we varied $\tau$ and $K$, and the results are presented in Figure 3. 1) It is observed that increasing temperature produces longer resoning and boosts recommendation performance, *i.e.,* , NDCG@5. A higher temperature introduces greater sampling entropy, allowing the model to explore a wider range of reasoning trajectories. 2) Conversely, it is found that increasing top-$K$ actually shortens the reasoning length and generally leads to a decline in recommendation performance. This is because a larger top-$K$ enlarges the candidate token set, which counterintuitively mitigates length hacking yet excessively large $K$ reintroduces noisy and low-quality samples.

Table 3: Comparison of sequence embedding extraction strategies.

| Method | N@5 | N@10 | H@5 | H@10 |
|---|---|---|---|---|
| Special token | 0.0353 | 0.0432 | 0.0509 | 0.0737 |
| Mean pooling | 0.0368 | 0.0453 | 0.0528 | 0.0796 |
| Max pooling | 0.0382 | 0.0445 | 0.0595 | 0.0729 |
| **Last hidden state (ours)** | **0.0388** | **0.0457** | **0.0588** | **0.0804** |

### 4.4.3 Analysis on Group Size

During RecPO, we sampled a group of trajectories to estimate their advantages. Therefore, we conducted experiments to analyze how varying group sizes impact performance. The results are outlined in Figure 3, and we gained several key findings as follows: 1) It is observed that performance improves for both backbones as group size increases, but the rate of improvement gradually slows down. While a larger group size generally leads to more explored paths, it also raises the training cost. These results suggest that selecting a group size of 6 or 8 is sufficient, and further increasing the group size is unnecessary. 2) Comparing the two backbones, we found a difference in sensitivity to group size. Qwen's performance at a group size of 1 lags significantly behind Gemma's, but it improves rapidly as the group size increases. Gemma performs well even with smaller groups, likely because its pretraining exposed it to a broader range of reasoning scenarios, thereby equipping it with stronger initial reasoning-for-recommendation capabilities.

### 4.4.4 Analysis on Embedding Strategy

We further examined different strategies for obtaining sequence-level embeddings. Specifically, we compared the last hidden state used in $R^2$ec against max pooling, mean pooling, and adding a special-token under identical settings. As reported in Table 3, the last hidden state yields the best overall performance. Special token strategies yield weaker performance in our setting, likely because we adopt LoRA fine-tuning and no large-scale optimization on the LLLM backbone; it is thus challenging for the model to effectively learn to generate and utilize newly introduced tokens for global sequence representation.

### 4.5 Analysis on Reasoning Behavior : How Does $R^2$ec Reason to Recommend?

How exactly does $R^2$ec transform raw behavioral descriptions into insightful recommendations? While the previous sections establish that reasoning substantially enhances recommendation accuracy, the underlying decision process—how the model internally reasons to reach its conclusions—remains less understood. To demystify this process, we conduct a systematic qualitative and quantitative analysis to uncover the reasoning strategies $R^2$ec employs across different domains and user contexts.

**Qualitative Analysis.**  We first engaged in extensive discussions with colleagues to collaboratively analyze and summarize the diverse reasoning behaviors exhibited by $R^2$ec. Through this process, we identified seven distinct reasoning strategies of $R^2$ec for recommendation:

1. Attribute Abstraction: Identifying underlying attributes of purchased items.
2. Pattern Recognition & Clustering: Detecting frequent item patterns to form user-interest clusters.
3. Scenario/Role-Based Reasoning: Inferring user roles or specific use cases.
4. Temporal Reasoning: Leveraging purchase timing and order for contextual understanding.
5. Self-Explanation: Providing explicit rationales for recommendations.
6. Negative Preference Exclusion: Avoiding previously negatively rated item types.
7. Multi-Objective Balancing: Managing trade-offs between conflicting user objectives.

These reasoning strategies illustrate how reasoning aids recommendation. Equipped with reasoning, large recommender models can distill salient user preferences from noisy or complex behavioral data, abstract beyond surface-level co-occurrence, and produce more contextually appropriate and interpretable recommendations.

**Quantitative Analysis.**  Building on the above findings, we then randomly sampled 100 reasoning outputs from each test set. These samples were annotated and analyzed through a combination of detailed human evaluation and automated assessment with GPT-4.1, ensuring both depth and consistency. Figure 4 visualizes the proportion of sampled reasoning outputs within each dataset that exhibit

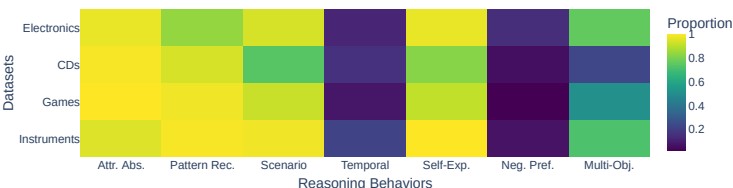

Figure 4: Distribution of reasoning behaviors across datasets. Each bar represents the proportion of reasoning outputs exhibiting a given reasoning behavior within a dataset.

each reasoning behavior, which reveals two key trends. (1) The distribution of reasoning behaviors varies across datasets, indicating that $R^2$ec self-adapts its reasoning patterns to domain characteristics and user–item interactions. (2) Even within a single domain, $R^2$ec flexibly selects different reasoning strategies depending on the specific user sequence. Overall, these results demonstrate that $R^2$ec does not follow a fixed reasoning template but instead employs context-aware reasoning tailored to both dataset and user, yielding improved recommendation performance and interpretability.

### 4.6 Analysis on Inference Efficiency

To quantify the efficiency gains of our unified model $R^2$ec over both traditional and LLM-based recommenders, we conducted comparative latency tests. All inferences were performed on a single NVIDIA RTX 3090 GPU using identical input prompts, maximum token lengths, and candidate sets. We executed 100 queries per run at a batch size of 1 and averaged the results over 3 independent runs.

Table 4: Average inference latency (in seconds) across models.

| Method | Latency (s) |
|---|---|
| SASRec | 0.014 |
| LangPTune | 1.90 |
| $D^3$ | 4.62 |
| LLaRA | 5.23 |
| $R^2$ec | **1.67** |
| $R^2$ec (with VLLM) | **0.0945** |

As shown in Table 4, 1) $R^2$ec achieves competitive inference efficiency among LLM-based recommenders, outperforming comparable methods such as LangPTune, $D^3$, and LLaRA. Such benefits stem from the use of scalable item embeddings, which avoid expensive autoregressive decoding over large item vocabularies and the design of an unified architecture. 2) When deployed with the VLLM inference framework, $R^2$ec significantly narrowed the efficiency gap with traditional sequential models like SASRec. These results demonstrate that $R^2$ec successfully balances expressive power and computational efficiency.

### 4.7 More Analysis

Due to space constraints, further extended analyses on *Scaling to Larger Backbones* and *Cross-Domain Robustness* are reported in Appendix I, while Appendix J provides representative *case studies*. We also discuss the *limitations* of our approach in Appendix H.

## 5 Conclusion and Future Work

In this study, we investigate the integration of reasoning into large recommender models by introducing $R^2$ec, a unified large recommender model with intrinsic reasoning capabilities, trained with RecPO, without reliance on human-annotated reasoning annotations. Extensive experiments and analysis demonstrated that $R^2$ecachieves both higher recommendation quality and lower inference latency compared to existing reasoning-augmented and conventional LLM-based recommenders. Taken together, these findings highlight the importance of tightly coupling reasoning and recommendation to unlock large recommender model's full potential. In the future, we aim to further investigate the efficient reasoning in larger recommender models, striving for optimal "thinking" in recommendations.

## Acknowledgments and Disclosure of Funding

The work described in this paper was supported by Research Grants Council of Hong Kong(PolyU/15209724, PolyU/15205325, PolyU/15207122), and PolyU internal grants (BDWP).

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

# A    Related Work

## A.1    Reinforcement Learning for LLM Reasoning

OpenAI's GPT-o1 demonstrated that scaling reinforcement learning (RL) training with large compute budgets enables models to internalize reasoning as a differentiable policy, achieving state-of-the-art performance in emergent meta-reasoning tasks [41]. Early RLHF methods trained a reward model on human preference data and fine-tuned the policy using Proximal Policy Optimization (PPO) [28]; however, PPO's multiple optimization passes introduce implementation complexity. To streamline RL tuning, Direct Preference Optimization (DPO) [42] performs single-step policy updates, trading simplicity for potential off-policy bias. Alternative estimators, such as Group Relative Policy Optimization (GRPO) [27], derive baselines from group-level rewards, eliminating the need for a separate critic, while REINFORCE's leave-one-out (RLOO) [31] computes advantages by excluding each trajectory sequentially. Building on these advancements, DeepSeek-Zero [40] removes reliance on supervised fine-tuning by driving reasoning emergence purely through intrinsic RL rewards, and DeepSeek-R1 further integrates rule-driven reward engineering with self-play verification to enhance reasoning robustness.

## A.2    Large Language Model-based Recommendation

Recommendation systems are a cornerstone of modern information services [43, 44], and the integration of Large Language Models (LLMs) has emerged as a promising frontier to advance their capabilities [45, 2, 46]. Recent advances in LLM-based recommendation typically follow two paradigms: 1) LLM-enhanced, where LLMs enrich recommendation pipelines with additional features, and 2) LLM-centric, where recommendation is framed as a generative task via item identifiers [1–4, 24–26], or LLMs serve as encoders to embed users and items [7, 29]. We term models under the second paradigm as large recommender models. In the first paradigm, LLM-generated embeddings are utilized to enhance conventional recommenders. Adoption of LLM-generated text are not new, typical examples includes user intent [16, 47], search queries [21], item summaries [19], or rationales [23]. Among them, SLIM [23], LangPTune [16], and Rec-R1 [21] further optimize LLM for finer rationale generation with RL. However, such designs introduce high resource demands for training and serving both models, and the connection between two modules remains gradient-less, blocking true end-to-end learning, resulting in suboptimal convergence and performance. Large recommender models typically recommend through autoregressive generation of item identifiers [4, 24] or titles [1–3, 25], or by embedding user interaction sequences [7, 29]. Large recommender model has demonstrated remarkable potential, but current methods fail to utilize reasoning ability of LLMs. To the best of our knowledge, no existing large recommender model improves recommendation over explicit reasoning generation, which is the gap that motivated our work.

# B    Dataset

Table 5: Dataset Statistics.

| Dataset | Users | Items | Density | Interactions | Train | Val | Test |
|---|---|---|---|---|---|---|---|
| Video Games | 29,230 | 10,144 | 0.031% | 63,502 | 50,801 | 6,350 | 6,351 |
| CDs and Vinyl | 7,701 | 12,024 | 0.023% | 13,435 | 10,748 | 1,343 | 1,344 |
| Musical Instruments | 15,656 | 10,320 | 0.031% | 34,373 | 27,498 | 3,437 | 3,438 |

Following the temporal-truncation protocol introduced in previous literature [3, 2], we construct three subsets—**CDs and Vinyl** (CDs), **Video Games** (Games), and **Musical Instruments** (Instruments) – from the lastest public Amazon review dataset[2] spanning from May 1996 to October 2023. For each domain, we begin with the most recent year of interactions (October 2022 – October 2023) and, if the resulting number of valid items is insufficient, iteratively roll the time window backward month-by-month until 10k items are obtained. We *omit* the 5-core filter so as to retain the nature behaviour characteristic of recommendation scenarios. Each user's interaction history is then chronologically sorted and truncated to the *latest 20 actions*, yielding fixed-length sequences for all subsequent

---

[2] https://amazon-reviews-2023.github.io/index.html.

modelling and evaluation. Finally the dataset is split with 80% training 10% validation 10% test. The resulting statistics are listed in Table 5.

## C    Baselines

- GRU4Rec [36] utilizes the GRU (Gated Recurrent Unit) architecture to model sequences.
- Caser [37] employs both horizontal and vertical convolutional operations to enhance the capture of high-order interactions within item sequences, improving recommendation accuracy.
- SASRec[38] incorporates a multi-head self-attention mechanism in its self-attentive sequential recommendation model, facilitating the modeling of intricate sequential data patterns.
- TIGER [9] learns extra tokens from item features to present items, and then converts the user sequence into the sequence of the new item token for next-item generation.
- BigRec [2] utilizes the item title to present the user sequence for recommendation generation.
- $D^3$ [3] proposed to address amplification bias and homogeneity issues in LLM recommenders that generate item title generation.
- SDPO [25] introduces a multi-negative, softmax DPO to better align LLMs with human preferences and mitigate biases in recommendation tasks.
- Llara [1] proposes aligning large language models with sequential recommenders through hybrid prompt design and curriculum learning for better sequential behavior understanding.
- SPRec [39] employs a self-play mechanism to iteratively debias preference alignment in LLM-based recommendations by suppressing undesirable items and reinforcing positive samples.
- LangPTune [16] utilized LLM as a user profile generator, and proposed an end-to-end training pipeline that optimizes LLM-generated user profiles for the recommendation objective.

## D    Implementation Details

- Hardware: 4x NVIDIA 3090 (24GB) GPUs
- Framework: PyTorch 2.5.0, Transformers 4.48.1, DeepSpeed 0.15.7

**Traditional Recommenders.**    We train all non-LLM baselines with the Cross-Entropy loss and the AdamW optimiser (learning-rate $1 \times 10^{-4}$); the batch size is 256. TIGER [9] adopts T5 as its backbone without further architectural changes.

**LLM-based Methods.**    For every LLM variant – including the contrastive-learning baseline (*CL*, Section 4.3) – we finetune `Gemma2-2B-Instruct` and `Qwen2.5-3B-Instruct`. Models are adapted with LoRA (rank = 4) using DeepSpeed Stage 2 and the quantized PagedAdamW optimizer. Training lasts for at most three epochs with early stopping (patience = 1); the maximum generation length is 512 tokens. If public implementations exist, we keep the original hyper-parameters; otherwise, a grid search over learning rates $\{5 \times 10^{-4}, 1 \times 10^{-4}\}$ is performed.

**R$^2$ec (ours).**    The learning rate is set as $1e-5$ with a batch size of 24, and apply linear warm-up for the first 32 steps. For trajectory generation, we set the group size $G$ as four, utilized vLLM[3] (tensor-parallelism = 1, target GPU utilisation = 95 %) for efficient generation. vLLM reserves one GPU for inference and leaves three for training. Sampling uses temperature = 1.5 and top-$K$ = 200. Policy optimization follows the clipping ratio $\epsilon = 0.2$ and omits KL regularization. Rewards are computed with NDCG@1000 and $\beta = 0.05$. Unless stated otherwise, these settings are used throughout.

## E    Overall Pipeline

**Training**    To instantiate R$^2$ec, given the dataset $D$, the initial policy $\pi_\theta$, and item embedding cache $\mathbf{H}_\mathcal{V}$, we train the model with detailed training process illustrated in Algorithm 1.

---

[3] https://docs.vllm.ai/en/latest/

**Algorithm 1** Training Process

---

**Input:** Dataset $\mathcal{D}$, initial policy $\pi_\theta$, embedding function $f_\theta$, item embedding table $\mathbf{H}_\mathcal{V}$
**Output:** Optimized policy model $\pi_\theta$
1: **for** step $= 1$ to $N$ **do**
2:     **if** step % $T_{\text{refresh}} == 0$ **then**
3:         Refresh item embedding: $\mathbf{H}_\mathcal{V}[v] \leftarrow f_\theta(x_v), \quad \forall v \in \mathcal{V}$
4:     **end if**
5:     Sample a training batch $\mathcal{B} = \{(u, v^+)\} \sim \mathcal{D}$
6:     Encode target item prompts and update embedding table: $\mathbf{H}_\mathcal{V}[v^+] \leftarrow f_\theta(x_{v^+}) \quad \forall(u, v^+) \in \mathcal{B}$
7:     **for all** $(u, v^+)$ in $\mathcal{B}$ **do**
8:         Generate $G$ trajectory: $\{[o_1, v^+]..., [o_G, v^+]\} \sim \pi_{\theta_{\text{old}}}(\cdot|x_u)$
9:         Compute reward for each trajectory using Eq. (5)
10:        Compute advantage for each trajectory using Eq. (2)
11:     **end for**
12:     Update policy parameters $\theta$ via loss in Eq. (8)
13:     Update old policy: $\theta_{\text{old}} \leftarrow \theta$
14: **end for**

---

**Inference** For inference, we first pass every item prompt $x_v$ through the trained model once to obtain all item embeddings $\mathbf{H}_\mathcal{V}$. At inference time, the model greedily generates a deterministic reasoning $o$ for a user prompt $x_u$, the last hidden state $\boldsymbol{h}_T$ goes through the embedding table, scoring each candidate via the inner product $s(v) = \boldsymbol{h}_T^\top \mathbf{H}_\mathcal{V}[v]$. The top-$K$ items can then be recommended by sorting these scores.

## F   Gradient Explanation for End-to-End Optimization

We explain how R$^2$ec achieves end-to-end optimization by analyzing gradient flow during training, where the recommendation signal propagates through both the reasoning trajectory and the live item encoding process.

During training, each item $v$ is encoded on the fly by the model as

$$h_v = f_\theta(x_v),$$

so that both the reasoning trajectory producing $h_T$ and the item embeddings $h_v$ depend on the same evolving computation rather than on static class weights.

At the recommendation step, the model predicts the ground-truth item $v^+$ via

$$p_\theta(v^+ \mid x_u, o_{1:T}) = \frac{\exp(h_T^\top h_{v^+}/\tau)}{\sum_{v' \in B} \exp(h_T^\top h_{v'}/\tau)},$$

and optimizes the policy objective for the highest-advantage trajectory $i^\star$:

$$\mathcal{L}_{\text{rec}}(\theta) = -A_{i^\star} \log p_\theta(v^+ \mid x_u, o_{1:T}^{(i^\star)}).$$

The corresponding gradient is

$$\nabla_\theta \log p_\theta(v^+) = \sum_{v' \in B} (\mathbf{1}[v' = v^+] - p_\theta(v')) \nabla_\theta s(v'), \quad s(v') = h_T^\top h_{v'},$$

which expands to

$$\nabla_\theta s(v') = (\nabla_\theta h_T)^\top h_{v'} + h_T^\top \nabla_\theta f_\theta(x_{v'}).$$

The second component, $h_T^\top \nabla_\theta f_\theta(x_{v'})$, shows that gradients flow directly into the item encoding process. Hence, updates from the recommendation signal not only refine the reasoning trajectory that produces $h_T$, but also reshape the item embedding function $f_\theta$, aligning the semantic encoding of items with the reasoning space.

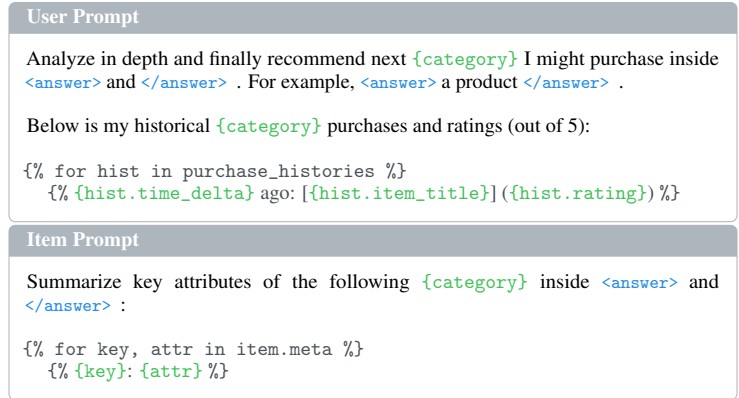

Figure 5: Prompt templates for user interaction history and item metadata. The *User Prompt* encodes a user's past purchases as a sequence of item titles, relative timestamps (e.g., "2hrs", "4d"), and explicit ratings (in $[1, 5]$), followed by an instruction to analyze and recommend the next item within the span of `</answer>` and `</answer>` . The *Item Prompt* summarizes structured item attributes (e.g., brand, type, features) with the same format requirement.

Table 6: Scaling analysis on Gemma backbones. Reasoning consistently improves recommendation performance across model sizes.

| Model | N@5 | N@10 | H@5 | H@10 |
|---|---|---|---|---|
| *Gemma-9B-it* | | | | |
| w/o reasoning | 0.0370 | 0.0460 | 0.0595 | 0.0893 |
| R$^2$ec | **0.0451** | **0.0527** | **0.0640** | **0.0878** |
| *Gemma-2B-it* | | | | |
| w/o reasoning | 0.0321 | 0.0393 | 0.0469 | 0.0692 |
| R$^2$ec | **0.0388** | **0.0457** | **0.0588** | **0.0800** |

## G  Prompt

As illustrated in Figure 5, user interaction histories and item metadata are serialized into token sequences via the given prompt templates. To signal the boundary between reasoning and recommendation, we introduce control symbol `<answer>` , which triggers a shift from language modeling head to recommendation head.

## H  Limitations

Our work acknowledges two primary limitations. First, introducing explicit reasoning generation inevitably increases inference latency and reduces efficiency due to additional autoregressive decoding steps; nevertheless, exploring the potential of reasoning within recommendation is valuable, and our experiments have empirically confirmed its effectiveness in improving recommendation accuracy. Second, constrained by computational resources, we employed parameter-efficient tuning (LoRA) rather than full-parameter fine-tuning, thus not fully demonstrating the potentially superior performance achievable through comprehensive optimization.

## I  Extended Analysis

**Scaling to Larger Backbones.**   We further scale R$^2$ec to **Gemma-9B-it**. The results in Table 6 confirm that reasoning consistently improves recommendation across model sizes, and the absolute gain grows with scale (21.7% higher N@5 than its non-reasoning counterpart).

**Cross-Domain Robustness.**   To further assess generalization to new domains, we extended our evaluation to the Electronics, MovieLens, and GoodReads datasets, and compared R$^2$ec with advanced baselines, as shown in Tables 6a, 6c, and 6b, respectively. R$^2$ec consistently surpasses prior methods,

Figure 6: Performance comparison across the Electronics (6a), GoodReads (6b), and MovieLens (6c) datasets. GoodReads and MovieLens evaluations follow the setting in SPRec [39] and SDPO [25], respectively, where baseline values are taken directly from their respective original papers.

<table>
<tr><td colspan="3" align="center">(a) Electronic</td></tr>
<tr><td>Method</td><td>N@5</td><td>H@5</td></tr>
<tr><td>GRU4Rec</td><td>0.0116</td><td>0.0157</td></tr>
<tr><td>Caser</td><td>0.0103</td><td>0.0164</td></tr>
<tr><td>SASRec</td><td>0.0113</td><td>0.0167</td></tr>
<tr><td>BigRec</td><td>0.0099</td><td>0.0146</td></tr>
<tr><td>$D^3$</td><td>0.0137</td><td>0.0214</td></tr>
<tr><td>LLaRA</td><td>0.0183</td><td>0.0279</td></tr>
<tr><td>$R^2$ec (ours)</td><td>**0.0205**</td><td>**0.0314**</td></tr>
</table>

<table>
<tr><td colspan="3" align="center">(b) GoodReads</td></tr>
<tr><td>Method</td><td>N@5</td><td>H@5</td></tr>
<tr><td>SASRec</td><td>0.0139</td><td>0.0202</td></tr>
<tr><td>BigRec</td><td>0.0236</td><td>0.0310</td></tr>
<tr><td>RW</td><td>0.0281</td><td>0.0380</td></tr>
<tr><td>$D^3$</td><td>0.0324</td><td>0.0413</td></tr>
<tr><td>SDPO</td><td>0.0315</td><td>0.0420</td></tr>
<tr><td>DMPO</td><td>0.0314</td><td>0.0410</td></tr>
<tr><td>SPRec</td><td>0.0330</td><td>0.0250</td></tr>
<tr><td>RosePO</td><td>0.0215</td><td>0.0300</td></tr>
<tr><td>$R^2$ec (ours)</td><td>**0.0354**</td><td>**0.0561**</td></tr>
</table>

<table>
<tr><td colspan="2" align="center">(c) MovieLens</td></tr>
<tr><td>Method</td><td>H@1</td></tr>
<tr><td>GRU4Rec</td><td>0.3750</td></tr>
<tr><td>Caser</td><td>0.3860</td></tr>
<tr><td>SASRec</td><td>0.3440</td></tr>
<tr><td>TALLRec</td><td>0.3895</td></tr>
<tr><td>MoRec</td><td>0.2822</td></tr>
<tr><td>ChatRec</td><td>0.2000</td></tr>
<tr><td>LLaRA</td><td>0.4730</td></tr>
<tr><td>SDPO</td><td>0.5260</td></tr>
<tr><td>$R^2$ec (ours)</td><td>**0.6211**</td></tr>
</table>

demonstrating the robustness of reasoning-enhanced recommendation across diverse domains and evaluation settings.

## J Analysis on Case Study

To illustrate domain-specific and backbone-specific reasoning behaviors, we analyze representative cases from the CDs and Video Games datasets (Figures 7, 8, 9, 10) using Gemma and Qwen as backbones. 1) On the CDs dataset, both models tend to reason by discussing several potentially interesting items and explicitly enumerating their features and genres, before providing a final recommendation. This behavior highlights the models' ability to leverage domain knowledge acquired during pretraining and reinforced through RecPO, for better recommendation. 2) On the Video Games dataset, the models shift towards inferring user intent from the purchase sequence, speculating on the likely category or intended use of the next purchase. This demonstrates a shift from feature-based enumeration to intent-driven reasoning as the domain changes. 3) Notably, Gemma generally produces shorter and more concise reasoning sequences compared to Qwen, which often generates more elaborate justifications. These patterns indicate that, while both models benefit from explicit reasoning, their generation style and reliance on pretrained knowledge can manifest differently across domains and backbone choices.

**Case Study 1 (CD and Vinyl)**

**User**

Analyze in depth and finally recommend next CD or vinyl I might purchase inside `<answer>`and `</answer>`. For example, `<answer>`a product`</answer>`.

Below is my historical CDs and vinyl purchases and ratings (out of 5):
255d 13.2h ago: [Pure 60's: The #1 Hits] (5.0)
255d 12.6h ago: [Seven] (4.0)
26.5min ago: [Comeblack] (4.0)
18.9min ago: [Hot Rocks 1964-1971] (5.0)
16.3min ago: [Return to Forever] (4.0)
9.4min ago: [Rock Believer] (5.0)
6.3min ago: [From Small Things: The Best of Dave Edmunds] (3.0)

$R^2$ec (Gemma2-2B)

Based on your past purchases, it seems you enjoy a mix of classic rock, soul/funk, and a touch of progressive rock. You appreciate well-crafted albums with strong vocals and a focus on musicality.
Here are some recommendations, keeping in mind your taste:
**CD/Vinyl:**

- **The Very Best of The Rolling Stones (CD/Vinyl):** You've enjoyed classic rock, and this compilation offers a great introduction to the Stones' iconic sound. It features their biggest hits and showcases their bluesy roots and rock 'n' roll energy.

- **The Dark Side of the Moon (Vinyl):** This Pink Floyd masterpiece is a classic for a reason. It's a complex and innovative album with a timeless sound. The vinyl format adds a special touch to the listening experience.

- **Kind of Blue (Vinyl):** Miles Davis's iconic jazz album is a must-have for any music lover. It's a beautiful and introspective album that showcases Davis's genius as a musician. The vinyl format brings out the nuances of the music.

- **The Miseducation of Lauryn Hill (CD/Vinyl):** This album is a powerful blend of hip-hop, soul, and R&B. Lauryn Hill's incredible vocals and the album's socially conscious lyrics make it a truly special listening experience.

**Why these recommendations:**

- **Variety:** These albums offer a mix of genres and styles, reflecting your diverse taste.

- **Quality:** They are all critically acclaimed and considered essential listening.

- **Format:** The vinyl format adds a unique and immersive experience, while the CD format offers convenience and portability.

Ultimately, the best way to find your next favorite album is to explore and discover new music. I recommend`<answer>`

Figure 7: Case Study on CDs with Gemma as backbone.

**User**

Analyze in depth and finally recommend next CD or vinyl I might purchase inside `<answer>`and `</answer>`. For example, `<answer>`a product`</answer>`.

Below is my historical CDs and vinyl purchases and ratings (out of 5): 97d 7.7h ago: [Live At The El Mocambo[2 CD]] (5.0)
97d 7.5h ago: [Radio Silence] (5.0)
2d 16.5h ago: [Little Box 'O' Snakes The Sunburst Years 1978-1982] (5.0)
2d 16.0h ago: [Goats Head Soup Super Deluxe] (5.0)

$R^2$ec (Qwen2.5-3B)

Given your history of purchasing and rating albums, it seems you have a strong preference for live recordings, classic rock, and deluxe editions. Based on this, here are some recommendations for CDs or vinyls that align with your tastes:

**1. "The Wall" (Deluxe Edition)**
**Artist:** Pink Floyd
**Format:** CD or Vinyl
**Description:** This is a legendary album that has been reissued multiple times, including a deluxe edition that includes rare tracks, documentaries, and bonus material. It's a must-have for any Pink Floyd fan.
**Rating:** Given your appreciation for deluxe editions and classic rock, this would likely receive a high rating.

**2. "The Dark Side of the Moon" (Super Deluxe Edition)**
**Artist:** Pink Floyd
**Format:** CD or Vinyl
**Description:** Another iconic album that has been extensively reissued. The super deluxe edition includes a wealth of bonus content, making it a comprehensive collection for fans.
**Rating:** Similar to "Goats Head Soup," this would also likely receive a high rating due to its extensive content and your appreciation for deluxe editions.

**3. "Live at the Royal Albert Hall" by David Bowie**
**Artist:** David Bowie
**Format:** CD or Vinyl
**Description:** This live album captures one of David Bowie's most famous performances, showcasing his versatility and the energy of his live shows. It's a great addition to your collection if you enjoy live recordings.
**Rating:** Given your enjoyment of live recordings, this would likely be highly rated.

**4. "The Complete Studio Albums" by Fleetwood Mac**
**Artist:** Fleetwood Mac
**Format:** CD or Vinyl
**Description:** This box set includes all of Fleetwood Mac's studio albums, providing a comprehensive overview of their career. It's a great way to explore their evolution as a band.
**Rating:** If you're a fan of Fleetwood Mac, this would likely be highly rated due to the quality of their music and the comprehensive nature of the collection.

**5. "The Best of the Beatles" (Deluxe Edition)**
**Artist:** The Beatles
**Format:** CD or Vinyl
**Description:** A compilation that includes some of the Beatles' most popular songs, arranged in chronological order. The deluxe edition often includes additional content like rare recordings and documentaries.
**Rating:** Given your appreciation for deluxe editions and classic rock, this would likely receive a high rating.

**Conclusion:**
Based on your past purchases and ratings, you might want to consider "The Wall" (Deluxe Edition), "The Dark Side of the Moon" (Super Deluxe Edition), "Live at the Royal Albert Hall" by David Bowie, "The Complete Studio Albums" by Fleetwood Mac, or "The Best of the Beatles" (Deluxe Edition). Each of these options offers a unique and comprehensive look into the artists' work, which aligns with your tastes.

Figure 8: Case Study on CDs with Qwen as backbone.

**Case Study 3 (Video Games)**

**User**

Analyze in depth and finally recommend next video game I might purchase inside `<answer>`and `</answer>`. For example, `<answer>`a product`</answer>`.

Below is my historical video games purchases and ratings (out of 5):

80d 1.6h ago: [For Apple Watch Fast Charger USB C, 2 in 1 Apple MFi Certified 3FT iWatch Charger Magnetic Charging Cable &PD USBC Fast Wall Charger Power Adapter for Apple Watch Series 6/SE/5/4/3/2/1&iPhone 14/13/12] (5.0)

76d 7.5h ago: [Kickstand Back Shell Holder Back Bracket for Nintendo Switch Console, Switch Replacement Kickstand Repair Kit with Screwdriver] (4.0)

45d 12.5h ago: [VOYEE Switch Controllers Wireless, Compatible with Switch Pro Controllers for Switch/Switch Lite/OLED/PC, One-Key Pairing Gamepad with 9-Color RGB LEDs 800mAh Battery 6-Axis Motion Wake Up Function] (5.0)

40d 23.5h ago: [Gvyugke 2.4GHz Wireless Gaming Headset for PS4, PS5, PC, Nintendo Switch, Bluetooth 5.2 Gaming Headphones with Mic for Mobile Device, Noise Canceling, Bass Surround, 50mm Driver, 40H Battery] (5.0)

$R^2$ec (`Gemma2-2B`)

It seems you enjoy tech accessories and gaming peripherals! Based on your recent purchases, you likely appreciate:

- **Functionality**: You're drawn to products that enhance your gaming experience, like the charging cable and the wireless gaming headset.
- **Convenience**: You value products that simplify your life, like the kickstand for your Switch.
- **Quality**:You prioritize products that are well-made and reliable, evidenced by your high ratings for the charging cable and the gaming headset.

Considering these preferences, here are some recommendations for your next video game purchase: `<answer>`

Figure 9: Case Study on Games with Gemma as backbone.

**User**

Analyze in depth and finally recommend next video game I might purchase inside `<answer>`and `</answer>`. For example, `<answer>`a product`</answer>`.

Below is my historical video games purchases and ratings (out of 5):

79d 15.5h ago: [BUSQUEDA Pro Head Strap with Battery for Quest 2, Magnetic Connection 6000mAh Replaceable VR Power, Lightweight & Balance Elite Strap Meta Accessories for Enhanced Support and Comfort in VR] (5.0)

69d 14.2h ago: [iTwano Stainless VR Stand for Quest 2 Pro, Desk Mount VR Holder and Headset Stand Compatible with Quest Pro 2 Rift S] (5.0)

59d 18.9h ago: [ROCCAT Vulcan II Max Optical-Mechanical PC Gaming Keyboard with Customizable RGB Illuminated Keys and Palm Rest, Titan II Smooth Linear Switches, Aluminum Plate, 100M Keystroke Durability Black] (5.0)

57d 21.9h ago: [ZIYOU LANG K3 Mechanical Gaming Keyboard 98 Keys with Number pad Chroma RGB Illuminated Water-Resistant Anti-Ghosting NKRO Retro Metal Panel Type-C for PC/Mac Gamer Typist Optical Clicky Blue Switch] (5.0)

57d 21.8h ago: [Corsair K60 PRO TKL Wired Optical-Mechanical OPX Linear Switch Gaming Keyboard with 8000Hz Polling Rate - Black] (4.0)

52d 16.8h ago: [IROK FE87/104 RGB Mechanical Keyboard, Hot Swappable Gaming Keyboard, Customizable Backlit, Magnet Upper Cover Type-C Wired Keyboard for Mac Windows-White/Brown Switch] (5.0)

**$R^2$ec (`Qwen2.5-3B`)**

Given your purchase history, it is clear that your interests focus primarily on VR setup accessories, gaming peripherals (particularly high-quality keyboards), and devices that enhance comfort and customizability during play, rather than on specific video games themselves.

Based on the trends observed in your history:

- **VR Content:** Multiple purchases of Quest 2/Pro accessories indicate a strong inclination towards immersive VR experiences.

- **Comfort & Ergonomics:** Investment in premium straps and ergonomic keyboards suggests that gaming comfort is a high priority.

- **Customizability and RGB Lighting:** The frequent selection of customizable RGB peripherals demonstrates your preference for visually dynamic and personalized setups.

With these points from your purchases, let's recommend one of the following types of next-gen games featuring improved immersive experiences for VR:

`<answer>`

Figure 10: Case Study on Games with Qwen as backbone.

