# OpenReview forum: "R$^2$ec: Towards Large Recommender Models with Reasoning"
_NeurIPS.cc/2025/Conference — NeurIPS 2025 poster_

### Official Review · Reviewer_idCa · 2025-06-20

**Clarity:** 3
**Significance:** 2
**Originality:** 2
**Rating:** 5
**Confidence:** 4

**Summary:**

This paper proposes a new method to leverage the capabilities of reasoning LLMs into the process of item recommendation. In more details, the paper introduces a method called R^2ec which consists of a reasoning LLM with an integrated recommendation head which allows for joint training of the "reasoning" and the "recommendation" components (i.e., combining generative and discriminative components into a single model).

The proposed method first describes the user interaction history into a prompt and is passed to an LLM. The final hidden state of the sequence generated by the LLM is then passed to the recommendation head, which performs an inner product with the embeddings of the items (obtained by feeding the textual description of the items through the LLM and using the last hidden state) to obtain a score for each item (higher score indicates the item is a better recommendation to the user). The model is trained using reinforcement learning (with a GRPO-like approach) with a reward composed of two components: one based on the NDCG ranking metric, and the other on a contrastive loss (promoting similarity with items in teh ground truth, and dissimilarity with other randomly sampled items).

The experimental section compares the proposed method with traditional and LLM-based baselines, and shows teh proposed method always achieves better results. A comprehensive set of ablation studies is performed to study the effects of the reasoning component, and on the stability and sensitivity of the reinforcement learning procedure based on the hyperparameters of the method.

**Questions:**

- It would be interesting to see how the method behaves in the cold start scenario (where there is no, or limited, history for a user/item). Have the authors performed any analysis of this?
- Could the authors please provide some information about the overhead runtime caused by the reasoning process?
- Could the authors mention why they only considered the latest 20 actions for each user? What happens when more actions are considered? Does the proposed method become too slow? How do the baselines scale (both in terms of runtime and performance) with longer history?
- Is the number of parameters of the models used for the proposed method comparable to that of the considered baselines?
- I've always found a bit strange the use of the embedding from the <EOS> token as a sentence embedding, as LLMs are not explicitly trained to put much information into this embedding. I understand that <EOS> has access to the whole sentence, but it still feels arbitrary. Have the authors tried different strategies to obtain the embeddings for a sequence (e.g., Mean/Max Pooling, or introducing special tokens)? The following papers may be of interest:
	- LLMs are Also Effective Embedding Models: An In-depth Overview, Tao et al.
	- LLM2Vec: Large Language Models Are Secretly Powerful Text Encoders, BehnamGhader et al.
	- Pooling And Attention: What Are Effective Designs For LLM-Based Embedding Models?, Tang et al.

**Ethical Concerns:**

["NO or VERY MINOR ethics concerns only"]

**Final Justification:**

The authors have addressed my concerns and added new experiments which I think will make the paper more valuable, therefore I raise my initial score

**Limitations:**

Yes, although I think it could be greatly improved by providing some numbers for the latency increase caused by the method

**Quality:**

3

**Strengths And Weaknesses:**

Strengths:
- The ablation study is very well done and studies in depth all the components of their method
- The quantitative results reported in the paper show the proposed method is significantly stronger than the considered competitors

Weaknesses:
- Reasoning models can be much slower than "non-reasoning" ones. This can be particularly problematic for real-world applications. Nevertheless, there is no discussion in the main paper about runtime. There is a limited discussion in the Appendix, but this could be greatly improved with some actual numbers
- The experiments shown in the paper are for a scenario with limited context (the authors say "Each user’s interaction history is then chronologically sorted and
truncated to the latest 20 actions"). This seems very far from real-world settings.
- There is no baseline with a simple pre-trained model that has received no fine-tuning/RL. I think adding such a baseline would be a great addition to the paper as it would really highlight the potential benefits of the proposed method.
- No discussion on the "cold-start" scenario.

---

> ### Author Rebuttal · Authors · 2025-07-31
>
> Thank you for your detailed and constructive review. We appreciate your recognition of our ablation study and empirical results. We acknowledge your concerns regarding inference latency, the limited user history setting, the absence of a pure pre-trained baseline, and the lack of cold-start analysis.
>
> We will address each of these points in detail in our response below.
>
> ---
>
> > Q1: Inference Latency and Runtime Analysis
> >
>
> We appreciate the reviewer’s question regarding the runtime overhead introduced by the reasoning process.
>
> First, we would like to clarify that the runtime of R²ec have been reported in Appendix G (“Analysis on Inference Efficiency”), the quantitative results on per-sample latency under different batch sizes.
>
> Second, to provide more detailed comparison and analysis, we have expanded our latency evaluation with additional benchmarks using consistent hardware (RTX 3090) and unified implementation settings. Below, we present the runtime (in seconds per sample) for R²ec and its VLLM serving version, against various baselines.
>
> |  | Runtime (s/sample) |
> | --- | --- |
> | LangPTune | 1.90 |
> | D³ | 4.62 |
> | LLara | 5.23 |
> | R²ec | 1.67 |
> | R²ec (with VLLM) | 0.0945 |
>
> These results demonstrate that R²ec achieves competitive efficiency among LLM-based recommenders, particularly when using modern inference frameworks. We attribute this efficiency in part to our use of scalable item embeddings, which avoids costly autoregressive token decoding over large item codebooks and enables fast retrieval within a unified architecture. We hope this additional analysis provides a comprehensive view of the reasoning overhead and helps clarify the practicality of our approach.
>
> ---
>
> > Q2: Short User History and Limited Context
> >
>
> Thank you for this insightful observation.
>
> First, truncating user histories to the most recent actions is a common practice in the recommendation literature, primarily due to the structure of public datasets and for fair comparison across baselines.
>
> Second, the LLM-based baselines all truncated sequences to 10 recent actions in their original experiment settings; in fact, we increased this window to 20 in our experiments to better retain semantic and sequential information, supporting stronger reasoning capabilities.
>
> Nevertheless, we fully acknowledge such setting does not fully capture the complexity of real-world recommendation scenarios, where user histories are often much longer and richer. Addressing long-context modeling remains an important and challenging direction for the field.
>
> Due to time and resource constraints, we were unable to reprocess all the datasets and conduct additional experiments with extended histories in this submission, but we do recognize this as an important aspect to explore in future work.
>
> ---
>
> > Q3: Adding Zero-shot Baseline
> >
>
> Thank you for this valuable suggestion. We agree that including a baseline using a pure pre-trained LLM—without any fine-tuning or RL—would help clarify the benefits of our approach.
>
> We now report results for this "zero-shot" baseline. As shown below, raw pre-trained LLMs achieve very low performance on recommendation tasks. This further highlights the significant gains achieved by our proposed training scheme.
>
> - Qwen
>
>
>     |  | N@5 | N@10 | H@5 | H@10 |
>     | --- | --- | --- | --- | --- |
>     | CDs | 0.000692 | 0.000692 | 0.00148 | 0.0022321 |
>     | Video Games | 0.00016886 | 0.00023456 | 0.00028342 | 0.00048811 |
>     | Instruments | 0. | 0.00034 | 0. | 0.000975 |
> - Gemma
>
>
>     |  | N@5 | N@10 | H@5 | H@10 |
>     | --- | --- | --- | --- | --- |
>     | CDs | 0.00064089 | 0.00090592 | 0.00148 | 0.0022321 |
>     | Video Games | 0.000317 | 0.00047956 | 0.00047237 | 0.00094473 |
>     | Instruments | 0. | 0.0001155 | 0. | 0.0003253 |
>
> ---
>
> > Q4: No Discussion on Cold-start Scenario
> >
>
> Thank you for highlighting the importance of the cold-start scenario. In response, we conducted additional analysis on both “cold” (unseen user or item) and “warm” (seen user/item) settings. Results (NDCG@10) are shown below:
>
> |  | CDs (cold) | CDs (warm) | Games (cold) | Games (warm) | Instruments (cold) | Instruments (warm) |
> | --- | --- | --- | --- | --- | --- | --- |
> | R²ec | 0.0306 | 0.0448 | 0.0264 | 0.0282 | 0.0168 | 0.0225 |
> | SASRec | 0.0057 | 0.0216 | 0.0150 | 0.0280 | 0.0080 	 | 0.0213 |
>
> As shown, R²ec consistently outperforms SASRec in both cold and warm scenarios, especially under the more challenging cold-start setting. This demonstrates the strong generalization and robustness of our method.
>
> ---
>
> > Q5: Model Parameter Comparability
> >
>
> For all LLM-based baselines, the parameter count is kept strictly consistent by adapting LORA finetuning to ensure fair comparison. For traditional (non-LLM) baselines, model sizes are inherently smaller by design; for clarity, we will provide a summary table of model sizes in the appendix.
>
> > Q6: Sequence Embedding Extraction Strategies
> >
>
> Thank you for raising this important point and for sharing relevant recent works. Your concern strongly resonates with my own initial explorations—in the early stages of this project, I experimented with several sequence embedding strategies, but I did not observe substantial or consistent improvements over the last hidden state, so I adopted it as the default for simplicity and robustness.
>
> Regard the papers. I found your recommended works extremely insightful—especially the discussion on enabling bidirectional encoding. Within the LLM + reasoning + embedding architecture, systematically investigating how to construct more effective embeddings is a highly promising direction for future research.
>
> In direct response to your suggestion, I have now re-evaluated these embedding strategies in controlled experiments. The results are as follows:
>
> |  | N@5 | N@10 | H@5 | H@10 |
> | --- | --- | --- | --- | --- |
> | last hidden state (ours) | 0.0388 | 0.0457 | 0.0588  | 0.0804 |
> | max pooling | 0.038158 | 0.044549 | 0.059524 | 0.072917 |
> | mean pooling | 0.036831 | 0.045287 | 0.052827 | 0.079637 |
> | special tokens | 0.035255 | 0.043157 | 0.050851 | 0.073661 |
>
> These results suggested that the last hidden state currently yields the strongest performance, while mean pooling improves training stability (though, due to rebuttal constraints, I cannot show the full curves here). Special token strategies yield weaker performance in our setting, likely because we adopt LoRA fine-tuning and no large-scale optimization on the LLLM backbone, it is thus challenging for the model to effectively learn to generate and utilize newly introduced tokens for global sequence representation.
>
> ---
>
> **Summary**
>
> We thank the reviewer again for the detailed feedback, which has prompted us to significantly strengthen our empirical analysis and clarify methodological choices. In the revised manuscript, we will incorporate all of the above additions—including expanded latency analysis, the zero-shot baselines, cold-start results, reporting of model sizes, and a systematic comparison of embedding strategies. We believe these enhancements address your concerns and further demonstrate the robustness and transparency of our approach.

---

> > ### Comment · Reviewer_idCa · 2025-08-02
> > **Reviewer Response**
> >
> > Thank you very much for the detailed answer and the new results. I believe my concerns are addressed and I am willing to raise my score

---

> > > ### Author Response · Authors · 2025-08-03
> > >
> > > Thank you again for your positive response. We greatly appreciate your time and effort, and should you have any further questions or suggestions, please feel free to reach out at any time.

---

### Official Review · Reviewer_ZdkM · 2025-06-27

**Clarity:** 2
**Significance:** 3
**Originality:** 2
**Rating:** 4
**Confidence:** 4

**Summary:**

This paper has explored combining the reasoning capabilities of LLMs into recommendations. The authors found that the existing related works often decouple the reasoning and recommendation process, leading to additional resource cost and suboptimal optimization. To address these problems, the authors devised two heads for one LLM to complete the reasoning and recommendation tasks, respectively. Then, using RL to optimize these two tasks jointly, they achieve a combination of reasoning and recommending. By extensive experiments on three datasets, the effectiveness of the proposed R2ec is validated.

**Questions:**

All my questions have been included in the weakness section.

**Ethical Concerns:**

["NO or VERY MINOR ethics concerns only"]

**Final Justification:**

The authors have addressed most of my concerns.

**Limitations:**

Yes

**Quality:**

3

**Strengths And Weaknesses:**

Strengths:
- S1. This paper is well-written and well-organized, making it easy to follow.
- S2. The code has been released.
- S3. Extensive experiments have been conducted.


Weaknesses:

- W1. The term "large recommender models" used in this paper is confusing. In general, "large recommender models" refers to a typical discriminative recommendation model with a large number of parameters, indicated in the previous works [1, 2]. By comparison, this paper has addressed the LLM as a recommender system.
- W2. Some notations are not clear. For example, in line 192, what does the reasoning token o represent? Is it the same as the token h in Figure 1?
- W3. The motivation of this paper is not really reasonable. The authors aim to combine and co-optimize the reasoning and recommending capabilities of LLMs. However, it is still unclear how the reasoning can benefit the final recommendation. Is it really necessary to maintain the reasoning procedure when recommending?
- W4. Table 1 shows that SASRec often outperforms TIGER and BIGRec, which cannot align with the original papers. I'm confused about it. I suggest that the authors analyze the reason or highlight the differences in experimental settings.
- W5. This paper belongs to LLMs as RS, but many important baselines have been ignored, such as LLaRA [3] and SPRec [4].

[1]. Actions Speak Louder than Words: Trillion-Parameter Sequential Transducers for Generative Recommendations

[2]. FuXi-𝛼: Scaling Recommendation Model with Feature Interaction Enhanced Transformer

[3]. Llara: Large language-recommendation assistant

[4]. SPRec: Leveraging Self-Play to Debias Preference Alignment for Large Language Model-based Recommendations

---

> ### Author Rebuttal · Authors · 2025-07-31
>
> We thank the reviewer for the positive and constructive feedback. We appreciate your recognition of  writing quality, code release, and experimental efforts.
>
> There are also some questions including the terms, notations, and baselines. We believe these issues could be clarified in the rebuttal stage, which we address as follow.
>
> ---
>
> > 1: On the use of the term "large recommender models"
>
> Thank you for highlighting this concern. We would like to clarify our terminology as follows.
>
> ### 1) Definition and technical equivalence.
> In prior literature [1,2], “large recommender models” generally refers to recommenders with large number of parameters, typically built on Transformer architectures. Although naming conventions differ, there is no fundamental technical gap between those models and ours; both rely on large-scale Transformer backbones, with the primary difference being whether the model is pretrained as a causal language model.
>
> ### 2) Motivation for terminology.
> We use “large recommender models” to emphasize our focus on unified, end-to-end LLM recommenders that do not require extra modules, in contrast to dual frameworks like LangPTune and SLIM, where LLM is merely the reasoner, but not the recommender(see Related Work). In this sense, the more commonly used term “LLM-based recommender” cannot fully capture this architectural distinction, as many so-called “LLM-based” systems still rely on external modules.
>
> Nevertheless, we appreciate your suggestion and will revise the manuscript to more clearly define this terminology in both the Introduction and Related Work sections, making explicit the architectural distinctions and ensuring our positioning is transparent to readers.
>
> ---
> > Q2: On notation clarity (reasoning token  $o$, hidden state $h$ )
>
> 1. The “reasoning token”   $o$ refers to the output token generated at each step during the model’s autoregressive reasoning process, as stated in line 192; and illustrated as purple blocks in Figure 1, as stated in the legend.
> 2. $h$ represents the model’s hidden states, not tokens, as stated in line 138-139, and illustrated in the legend of Figure 1.
> 3. To further improve clarity, we will revise the figure by adding a prominent arrow to clearly indicate the correspondence between the reasoning tokens and the illustration.
>
> ---
> > Q3: On the motivation and necessity of reasoning in recommendation
>
> Thank you for raising this insightful question regarding the necessity of explicit reasoning in recommendation. We will derive this question into the following 3 parts, namely the research necessity, the empirical evidences, and the Reasoning Behavior Analysis.
>
> ### 1) Motivation and Research Necessity.
> Generative recommendation (GR) built upon LLM backbones has already shown strong promise in both industry and academia. While recent advances demonstrate that reasoning is central to LLMs’ performance in complex tasks such as mathematics. Given that recommendation inherently involves intuitive reasoning—such as inferring user intent and abstracting preferences — it is both natural and timely to investigate whether reasoning can yield tangible gains in GR setting. We believe exploring this direction is crucial for advancing the capabilities and practical impact of GR  systems.
>
> ### 2) Empirical Evidences.
> 1. Ablation studies (Section 5.3) demonstrated that our model consistently outperforms a variant trained solely with contrastive loss and *without* reasoning. This provides quantitative evidence that incorporating reasoning tokens leads to better recommendation performance.
>
> 2. In direct response to “is it really necessary to maintain the reasoning procedure”, we performed an additional ablation where R²ec is forced to output recommendations *without* generating any reasoning (denoted as w/o r), comparing with the reasoning one (denoted as w r). As shown below, removing reasonings consistently degrades performance across all domains.
>
> 	||**N@5**|**N@10**|**H@5**|**H@10**|
> 	|-|-|-|-|-|
> 	|CDs w/o r|0.0222|0.0277|0.0327|0.0469|
> 	|CDs w r|0.0388|0.0457|0.0588|0.0804|
> 	|Video Games w/o r|0.0145|0.0184|0.0220|0.0341|
> 	|Video Games w r|0.0205|0.0271|0.0288|0.0532|
> 	|Instruments w/o r|0.0156|0.0178|0.0224|0.0309|
> 	|Instruments w r|0.0161|0.0203|0.0264|0.0397|
> 	|Electronics w/o r|0.0169|0.02160|0.0265|0.0414|
> 	|Electronics w r|0.0205|0.0252|0.0314|0.0462|
>
> These experiments directly confirm that **explicit reasoning is not only useful, but necessary for maximizing the performance of generative recommendation models**.
>
> ### 3) Reasoning Behavior Analysis.
> To directly address the question of “how reasoning benefits the final recommendation, "we conducted an in-depth both qualitative and quantitative analysis of R²ec's reasoning patterns.
>
> We first engaged in extensive discussions with colleagues to collaboratively analyze and summarize the diverse reasoning behaviors exhibited by R²ec. Through this process, we identified 7 typical reasoning strategies. Building on this framework, we then randomly sampled 100 reasoning outputs from each test set. These samples were annotated and analyzed through a combination of detailed human evaluation and automated assessment with GPT-4.1, ensuring both depth and consistency in our analysis.
>
> 1. **Qualitative Analysis.** We found that there are 7 different reasoning behaviors for recommendation, including:
>
> 	- Attribute Abstraction: Identifying underlying attributes of purchased items.
> 	- Pattern Clustering: Detecting frequent item patterns to form user-interest clusters.
> 	- Role-Based Reasoning: Inferring user roles or specific use-cases.
> 	- Temporal Reasoning: Using purchase timing and order for context.
> 	- Self-Explanation: Providing explicit rationales for recommendations.
> 	- Negative Preference Exclusion: Avoiding previously negatively-rated item types.
> 	- Multi-Objective Balancing: Managing trade-offs between conflicting user objectives.
>
> 2. **Quantitative Analysis.** To further substantiate our findings, we quantified the distribution of reasoning behaviors across different datasets. As noted, each reasoning output may exhibit multiple behaviors simultaneously. The table below reports the **proportion** of sampled reasoning outputs within each dataset that display each behavior:
> 	|Dataset|Attribute Abstraction|Pattern Clustering|Role-Based Reasoning|Temporal Reasoning|Self-Explanation|Negative Preference Exclusion|Multi-Objective Balancing|
> 	|-|-|-|-|-|-|-|-|
> 	|Instruments|0.95|0.99        |0.98     |0.22       |1.00|0.07     |0.72  |
> 	|Games|1.00|0.98        |0.92     |0.08       |0.91|0.02     |0.51  |
> 	|CDs|0.99|0.94        |0.74     |0.16       |0.82|0.06     |0.23  |
> 	|Electronics|0.97|0.84        |0.94     |0.12       |0.97|0.15     |0.76  |
>
> 	Note: Each value reflects the proportion of sampled reasoning outputs within the corresponding dataset that demonstrate a given reasoning behavior.
>
> 	This quantitative results highlight several key trends. First, the distribution of reasoning behaviors differs across datasets, indicating that after training, the model self-adapts its reasoning patterns to fit the characteristics of each domain and its user-item interactions. Second, the variability in these proportions suggests that even within a single domain, R²ec flexibly selects different reasoning strategies based on the specific user sequence.
>
> Overall, these results demonstrate that the model employs context-aware reasoning tailored to both the dataset and the individual user, contributing to improved recommendation performance and ultimately proving its necessity.
>
> ---
> > Q4: On SASRec performance and discrepancies with original papers
>
> Thank you for pointing out this discrepancy. As described in Appendix C, our SASRec implementation uses cross-entropy (softmax) loss instead of binary cross-entropy (BCE), which significantly boost performance. We highlight that such findings are reasonable and consistent with previous findings [1], which reported that softmax-based SASRec can outperform many larger recommendation models. We do value your concerns and will highlight this implementation detail more clearly in the main text to avoid confusion.
>
> ---
> > Q5: On missing baselines
>
> Thank you for highlighting the importance of recent works such as LLaRA[2] and SPRec[3].
>
> To ensure broader coverage of recent advances, we supplemented our original baselines with three additional LLM‑based recommenders—SPRec, SDPO, and LLara. We note that LLaRA and SDPO are originally designed as rerankers in a candidate‑based setting, thus not comparable to our end‑to‑end generation framework. To provide fair comparison, we have adapted their implementations by removing the candidates prompts and instead performing **constrained generation over the full item corpus**, denoted as LLaRA* and SDPO*.
>
> The updated comparison is shown below:
> ||N@5|N@10|H@5|H@10|
> |-|-|-|-|-|
> |SPRec|0.0022|0.00246|0.0029|0.0037|
> |SDPO*|0.0018|0.00273|0.00223|0.00521|
> |Llara*|0.0039|0.00488|0.009673|0.01265|
> |BigRec|0.0025|0.0039|0.0045|0.0089|
> |D^3|0.0057|0.0076|0.0082|0.0141|
> |ours|0.0388|0.0457|0.0588|0.0804|
>
> We hope these additions and clarifications adequately address your concern.
>
> ---
> In summary, we sincerely thank the reviewer for the constructive feedback. We greatly appreciate your recognition of our writing, code release, and experiments, as well as your thoughtful suggestions on terminology, notation, and baselines. We have carefully addressed each concern,, providing detailed clarifications, new analyses, and additional experiments where possible. We hope these responses further strengthen the value and validity of our work.
>
>
> [1] Are LLM-Based Recommenders Already the Best? Simple Scaled Cross-Entropy Unleashes the Potential of Traditional Sequential Recommenders
> [2] LLaRA: Large Language-Recommendation Assistant
> [3] SPRec: Self-Play to Debias LLM-Based Recommendation
> [4] On Softmax Direct Preference Optimization for Recommendation

---

> > ### Comment · Reviewer_ZdkM · 2025-08-04
> >
> > Thanks for your response, which has addressed most of my concerns. I will raise my score.

---

> > > ### Author Response · Authors · 2025-08-04
> > >
> > > Thank you for your encouraging feedback and for raising your score. We truly value your time and insights. Please feel free to contact us with any further questions or suggestions.

---

> ### Author Response · Authors · 2025-08-03
>
> Hi Reviewer ZdkM, if there are any points where our responses are unclear or have not fully addressed your concerns, we would be extremely grateful if you could let us know. This will allow us time to further clarify or revise our work accordingly. Your feedback is truly important to us, and we deeply appreciate your time, effort, and guidance.
>
> Thank you again for your consideration.

---

### Official Review · Reviewer_jU2r · 2025-06-30

**Clarity:** 4
**Significance:** 3
**Originality:** 2
**Rating:** 3
**Confidence:** 5

**Summary:**

Motiveted by the success of deep thinking in LLMs like DeepSeek R1, this paper explores this question: how can LLM benefit from reasoning to enhance recommendation performance?

The main challenge lies in the lack of annotated reasoning traces in the recommendation domains. To address this, the authors propose a fused reward which combines discrete ranking rewards with continuous similarity rewards to guide the RL training. Experiments on three datasets demonstrated superiority of the proposed model in terms of recommendation accuracy.

**Questions:**

Please see the weakness section.

One more thing: while I list "Experiments on three datasets demonstrate the proposed model outperform existing baselines with a significant high margin" as one strength, I observe the improvement of the proposed method is surprisingly high, e.g, in Video Games, improvements can be as high as 104%. According to my years of experience on recommender systems, either there are some problems in your experiments, or your paper is a significant breakthrough which deserves a nomination for best paper. Personally, I tend to take the former hypothesis. How can authors prove that the experiment setting is not problematic? I suggest you can use some scandalized benchmark datasets, in which metrics can be directly aligned with existing papers.

**Ethical Concerns:**

["NO or VERY MINOR ethics concerns only"]

**Final Justification:**

After considering the authors’ rebuttal and the discussion, my main concerns remain. The novelty is limited, as the technical contributions closely follow existing literature, and the motivation for using LLM-based reasoning in recommendation is not convincingly demonstrated. The latency analysis is unclear and does not reflect real-world serving conditions, and the comparison with baselines is neither comprehensive nor standardized, making it hard to judge the true effectiveness of the approach. While the paper is well written and the research question is interesting, the empirical results and justifications are not sufficient to outweigh these issues. For these reasons, I cannot recommend acceptance at this time, but I encourage the authors to strengthen their experimental comparisons, provide clearer motivation, and address latency concerns in future revisions.

====
updated in the second revision:
I have raised the initial score as authors provided additional experimental results which hit one of my questions.

**Limitations:**

See the weaknesses part.

Meanwhile, the authors also discuss the limitations in the appendix, but the latency discussion is not solid. More analysis on traditional recommender models, such as SASRec, is expected.

**Quality:**

2

**Strengths And Weaknesses:**

Strengths:
- The paper is well written and easy to follow.
- The studied problem--training reasoning-based LLM4Rec--is important and interesting.
- Experiments on three datasets demonstrate the proposed model outperform existing baselines with a significant high margin.

Weakness:
- The novelty of the paper is limited. Both adding classification heads to LMs and RL training pipeline is well setup in existing literature.
- While the studied research question of this paper:  "how can LLM benefit from reasoning to enhance recommendation performance? " , is interesting, the authors failed to deep dive and answer the question in a insightful and systematic way. The authors simply demonstrate by quantitative metrics on recommendation metrics. However, I am more interested to know some fundamental questions:  (1) why reasoning process matter so much for item recommendations? Test-time scaling works effectively for complex problems like Mathematical Olympiad problems and high-order logical tasks, but does item recommendation task involve this complex reasoning nature? (2) Explainable recommender models in literature, such as P5, mainly generate reasons to be displayed to users. But in this paper, the reasons seems not to display to users, but help boost the accuracy of recommender models.
- Recommender systems usually require real-time serving. The proposed large reasoning recommender LMs may not satisfy the requirement.
- Baselines are not enough: nowadays there are a lot of papers discussing leveraging LLMs for recommendations. The authors are suggested to compare with more baselines to make the experiments more convincing. I know there will never be a full list of adequate baselines in this direction, but at least the three baselines in the paper, BigRec, D^3, LangPTune, are far from representing the direction.

---

> ### Author Rebuttal · Authors · 2025-07-31
>
> Thank you for the thoughtful evaluation. We appreciate your recognition of the clarity, the significance of our research problem, and the strength of results.
>
> We would like to address several misunderstandings on our contributions. Specifically, the novelty, the role of reasoning in recommendation, latency, and experiment result.
> Below, we provide a detailed response to your 5 primary questions.
>
> > Q1: Novelty.
>
> While LLM reasoning advances make its adaptation to recommendation appear straightforward, our results show that **simply applying standard RL and reasoning approaches fails in practice** and critical design choices must be made. We summarize key contributions as **(1) model architect** and __(2) training method__.
>
> 1. Not a Classification Head — Architect considerations to couple reasoning and recommendation
>
> 	- Dual architect is suboptimal. Prior work typically adopts LLM as reasoner with recommender. This modular approach increases computation and decouples reasoning from recommendation, limiting effectiveness (see lines 37–47). Our goal is to design a unified architect that jointly accomplishes generative reasoning and discriminative recommendation.
> 	- Dense Embedding over Discrete Codebook. Generative recommendation (GR) often predict items as tokens from codebook, which restricts scalability, generalization to new items, and adds decoding cost. Since reasoning generation is already time-consuming, we use a dense item embedding paradigm for scalability and efficiency.
> 	- Item Semantics is crucial. While the simplest way to optimize dense embeddings in LLM is through a "classification head" (denoted as ClsH). Our experiments show that such design, with or without reasoning, performs poorly, under standard baselines like BigRec:
>
> 		|  | N@5    | N@10   | H@5    | H@10   |
> 		| ----- | ------ | ------ | ------ | ------ |
> 		| ClsH w/o reasoning  | 0.0017 | 0.0019 | 0.0022 | 0.0037 |
> 		| ClsH with reasoning | 0.0025 | 0.0027 | 0.0030 | 0.0045 |
> 		| BigRec| 0.0025 | 0.0037 | 0.0045 | 0.0089 |
>
> 		**Summary:** After extensive early-stage exploration and careful considerations, our final architect (i) tightly couples reasoning and rec in a unified model, (ii) leverages scalable, efficient dense item representations, and (iii) enables the model to exploit and update item semantics throughout training.
>
> 2. **RL** for Rec is Nontrivial and Requires New Solutions
>
> 	- Reasoning Supervision is Practically Impossible.
> 		Annotated reasonings are feasible to collect at scale for math / code tasks. In recommendation, however, user-level annotated reasoning is impossible to obtain due to the diversity and subjectivity of user intents. RecPO is a **fully self-supervised RL paradigm**, learning reasoning purely from interaction logs—an essential, but highly challenging adaptation for this domain.
>
> 	- Reward Design is Nontrivial.
> 		NLP domains benefit from clear reward signals (e.g., corectness). In recommendation, ranking metrics are the only available rewards. We found that directly using such metrics leads to weak result, we thus proposed **fused reward scheme** to combine discrete ranking signals with continuous similarity (Sec. 4.2.2, Table 2), leading to much more effective and stable training.
>
> 	**Summary:** RecPO is the first to enable **fully joint RL optimization** of both reasoning and recommendation without reasoning annotation. By introducing encoding and utilization of item semantics to training objective, and optimizing with fused reward, our approach tightly couples reasoning and ranking, allowing reasoning to directly enhance recommendation.
>
> > Q2: Why reasoning process matters?
>
> We agree that the reasoning of R²ec differs from mathematics domains. We thus conducted analyses regarding R²ec reasoning behaviors. By qualitatively annotating and clustering reasoning outputs, we found that R²ec typically leverages 7 reasoning strategies. To further substantiate the findings, we quantified the distribution of behaviors across different datasets as follow:
>
> |             | Attribute Abstraction | Pattern Clustering | Role-Based Reasoning | Temporal Reasoning | Self-Explanation | Negative Preference Exclusion | Multi-Objective Balancing |
> | ----------- | --------------------- | ------------------ | -------------------- | ------------------ | ---------------- | ----------------------------- | ------------------------- |
> | Instruments | 0.95                  | 0.99               | 0.98                 | 0.22               | 1.00             | 0.07                          | 0.72                      |
> | Games       | 1.00                  | 0.98               | 0.92                 | 0.08               | 0.91             | 0.02                          | 0.51                      |
> | CDs         | 0.99                  | 0.94               | 0.74                 | 0.16               | 0.82             | 0.06                          | 0.23                      |
> | Electronics | 0.97                  | 0.84               | 0.94                 | 0.12               | 0.97             | 0.15                          | 0.76                      |
>
> (Each value indicates the proportion of outputs exhibiting the behavior. Each output may contain multiple behaviors.)
>
> This analysis demonstrates that R²ec improves recommendation quality by leveraging diverse and adaptive reasoning strategies, enabling it to capture richer user intent and item semantics beyond shallow co-occurrence patterns.
>
> > Q3 Real-time serving and latency.
>
> We thank the reviewer for emphasizing real-time serving and latency concerns.
>
> To address this, we conducted comprehensive latency evaluation, including comparison with both LLM-based Rec baselines and traditional recommenders, summarized as below.
>
> |                  | latency |
> | ---------------- | ------- |
> | SASRec           | 0.014   |
> | LangPTune        | 1.90    |
> | D³               | 4.62    |
> | LLara            | 5.23    |
> | R²ec             | 1.67    |
> | R²ec (with VLLM) | 0.0945  |
>
> These results demonstrate that R²ec achieves competitive efficiency among LLM-based recommenders, particularly when using modern inference frameworks. We attribute this efficiency in part to our use of scalable item embeddings, which avoids costly autoregressive token decoding over large item codebooks and enables fast retrieval within a unified architecture. We hope this additional analysis provides a comprehensive view of the reasoning overhead and helps clarify the practicality of our approach.
>
> While R²ec is slower than lightweight models like SASRec, its unified reasoning–recommendation design brings substantial gains in recommendation quality, and aligns with the community’s ongoing shift toward more capable, explainable models. We believe such advances justify the increased computation and pave the way for efficient, practical LLM-based recommenders as the field continues to optimize for both performance and speed.
>
> > Q4: Baselines.
>
> We included additional LLM-based baselines — SPRec, SDPO, and LLara. Since LLaRA and SDPO are originally designed as rerankers, , we adapted implementations by removing candidates prompts with constrained generation over item corpus, denoted as LLaRA* and SDPO*. The updated result is shown below:
>
> |        | N@5    | N@10   | H@5    | H@10   |
> | ------ | ------ | ------ | ------ | ------ |
> | SPRec  | 0.0022 | 0.0025 | 0.0029 | 0.0037 |
> | SDPO*  | 0.0018 | 0.0027 | 0.0022 | 0.0052 |
> | Llara* | 0.0039 | 0.0049 | 0.0097 | 0.0127 |
> | BigRec | 0.0025 | 0.0039 | 0.0045 | 0.0089 |
> | D³     | 0.0057 | 0.0076 | 0.0082 | 0.0141 |
> | ours   | 0.0388 | 0.0457 | 0.0588 | 0.0804 |
>
> As illustrates, R²ec maintains a substantial lead over the expanded set, further validating its effectiveness.
>
> > Q5: Concerns on experiment result.
>
> Thank you for raising this important point and for your careful review. We take reproducibility and reliability very seriously, and address your concerns as follows:
>
> 1. Dataset Preprocessing.
> 	Our datasets are preprocessed following the D³ methodology (see Appendix for details), but without additional 5-core filtering. This results in a much higher proportion of cold-start cases—e.g., 39.9% of users in the CDs dataset and 34.7% in Instruments are cold-start. In such settings, generative baselines typically struggle, whereas R²ec benefits from richer semantics and demonstrates amplified improvements.
>
> 2. Baseline Implementation.
> 	For baselines such as LLaRA, BigRec, and D³, we strictly follow the original papers and use only the item titles as input features. This limited information further hinders their performance on cold-start items. In addtion, we reported relative improvements ((R²ec - baseline) / baseline), which can look inflated when baseline values are low.
>
> 3. Controlled Comparison
> 	Following your suggestion, we compared all methods using the same backbone (Gemma-2B) and LoRA adapter on the 2018 5-core CDs dataset. As shown below, R²ec consistently outperforms the version without reasoning (“w/o r”).
>
> |        | N@5    | N@10   | H@5    | H@10   |
> | ------ | ------ | ------ | ------ | ------ |
> | BigRec | 0.0384 | 0.0412 | 0.0449 | 0.0530 |
> | w/o r  | 0.0328 | 0.0384 | 0.0465 | 0.0637 |
> | R²ec   | 0.0386 | 0.0446 | 0.0535 | 0.0726 |
>
> This result aligns with our expectations and prior findings: as user sequences become more informative, the relative advantage of reasoning diminishes, but the positive effect of reasoning remains robust and measurable.
>
> 4. Commitment to Reproducibility.
> 	We have provided our full codebase in supplementary materials to enable the community to verify and build on our work.
>
> We hope these steps demonstrate the validity of our findings.

---

> > ### Comment · Reviewer_jU2r · 2025-08-04
> >
> > Thank you for the detailed response. Unfortunately, my major concerns remain, including the novelty, necessity of reasoning for recommender systems, latency, and gains over traditional recommenders.
> >
> > For example, there is no annotations for what the numbers in the latency table mean and what is the running environment/configurations. To evaluate the real-time latency to simulate online serving, you need to record the time-to-finish cost for inferring an individual user's request (which means the batch size should set to 1).  It is allowed to use KV cache and ignore the prefilling time, and only count the generation time cost.
> >
> > I am still not convinced by the responses in "Q2: Why reasoning process matters" and I think those aspects can be well captured by non-LLM recommender methods such as GNN-based models and multi-objective-aware models, since in the field of recommender systems, a lot of papers are proposed to tune model structures to implement these purposes.
> >
> > While I have strong concerns on the motivations, limitations and novelty of the paper, what I value most is the performance of the method. Personally, if a paper provides impressive results, I will ignore the technical novelty and other concerns like latency. However, the experiments in the current status cannot convince me.  I will suggest the authors to use a common setting for benchmarking methods that can be directly aligned with some existing papers' results, so that the public community can easily judge and verify whether the performance of baselines like SASRec is well represented (as my experience of reproducing research papers, many of them underrate baselines' results).

---

> > > ### Author Response · Authors · 2025-08-05
> > >
> > > > Q1: Novelty Concerns remain
> > >
> > > We have summarized contributions in previous response, specifically:
> > > - Unified design: A tightly integrated architecture adopting dense embeddings paradigm.
> > > - Reasoning optimization without labels: RecPO enables RL training with no need for annotated reasoning traces.
> > > - Reward scheme: a fused ranking + embedding similarity reward enables effective optimization.
> > >
> > > We understand the taste of novelty vary across individuals, but we would appreciate it if you could specify your concerns based on the rebuttal, as none were stated in your response.
> > >
> > > > Q2: Latency evaluation setting is not provided
> > >
> > > The latency evaluations were conducted as you stated, under batch size = 1,  with KV cache, without prefilling, and on identical hardware, in accordance with the setting detailed in Appendix G.
> > > The omission of setting is due to characters constraints,  we will ensure this is provided in the final version.
> > >
> > > > Q3: Why reasoning matters with the concern of non-LLM recommenders?
> > >
> > > You stated that "*the aspects of reasoning can be addressed by non-LLM recommenders*" and, “since *previous works have modified model structures to implement these purposes*,” the strategies in R²ec’s reasoning process do not justify the benefit of reasoning.
> > > We respectfully disagree such statement, and wish to clarify several key points:
> > > 1. LLM-based Recommendation is a promising direction.
> > >     Powered by massive pretraining and billions of parameters, LLM-based recommenders  with its generalization and semantic understanding abilities, have demonstrated significant potential, outperformed traditional models in many recent studies [1-8].
> > > 2. Reasoning is a validated useful extension for recommendation.
> > >     - Numerous works [5-9] have shown that reasoning boosts recommendation effectiveness. Empirical evidence suggests reasoning enables the model to infer deeper user preferences and motives by leveraging factual knowledge, intent, and external events—capabilities beyond what conventional models can do [5-7].
> > >     - Fundamentally, reasoning provides performance gains by introducing additional test-time compute [10], allowing iteratively refinements and generate more accurate result.
> > > 3. Our work builds on above context, with its own innovations.
> > >     While we do not claim to be the first to introduce reasoning into recommendation, our contribution is still significant: (a) a unified architecture, (b) an RL training algorithm, and (c) a fused reward scheme.
> > >     Our goal is to unify the rec+reasoning paradigm, achieving both effectiveness and conceptual elegance.
> > > 4. The fact that non-LLM recommenders can be enhanced by various strategies does not invalidate reasoning for LLM-based models.
> > >     While non-LLM recommenders advanced by architect modifications, this does not make it incorrect that reasoning can enhance LLM recommenders. These are two distinct research directions.
> > >
> > > > Q4: Benchmarking performance
> > >
> > > As no single “benchmarking” dataset is shared by all baselines, we conducted evaluations on two additional datasets and settings exactly as in the original papers, ensuring direct alignment with existing works:
> > > - **MovieLens**: from both LLaRA[2] and SDPO[3] (ranking within 20 candidates).
> > > - **GoodReads**: from SPRec[4].
> > >
> > > |MovieLens|H@1|
> > > |-|-|
> > > |GRU4Rec|0.375|
> > > |Caser|0.386|
> > > |SASRec|0.344|
> > > |TALLRec|0.3895|
> > > |MoRec|0.2822|
> > > |ChatRec|0.2|
> > > |Llara|0.473|
> > > |SDPO|0.526|
> > > |R²ec|**0.62105**|
> > >
> > > |GoodReads|N@5|H@5|
> > > |-|-|-|
> > > |SASRec|0.0139|0.0202|
> > > |BigRec|0.0236|0.0310|
> > > |RW|0.0281|0.0380|
> > > |D³|0.0324|0.0413|
> > > |SDPO|0.0315|0.0420|
> > > |DMPO|0.0314|0.0410|
> > > |SPRec|0.0330|00250|
> > > |RosePO|0.0215|0.0300|
> > > |R²ec|**0.0354**|**0.0561**|
> > >
> > > Note: all numbers except ours were taken verbatim from SDPO [3] and SPRec [4].
> > >
> > > Our empirical results show that R²ec outperforms baselines across different settings, validating its effectiveness.
> > >
> > > ---
> > > [1] “A Survey on Large Language Models for Recommendation.” *World Wide Web*
> > > [2] “LLaRA: Large Language-Recommendation Assistant.” *Proceedings of the 47th International ACM SIGIR Conference on Research and Development in Information Retrieval*
> > > [3] ‘On Softmax Direct Preference Optimization for Recommendation’. *Proceedings of the 38th International Conference on Neural Information Processing Systems*
> > > [4] “SPRec: Self-Play to Debias LLM-Based Recommendation.” April 28, 2025, 5075–84.
> > > [5]  “Can Small Language Models Be Good Reasoners for Sequential Recommendation?” *Proceedings of the ACM Web Conference 2024*
> > > [6] “Towards Open-World Recommendation with Knowledge Augmentation from Large Language Models.” *18th ACM Conference on Recommender Systems*
> > > [7] “Leveraging LLM Reasoning Enhances Personalized Recommender Systems.” *Findings of ACL 2024*
> > > [8] “ReLand: Integrating Large Language Models’ Insights into Industrial Recommenders via a Controllable Reasoning Pool.” ACM
> > > [9] “Scaling LLM Test-Time Compute Optimally Can Be More Effective than Scaling Model Parameters“. *The Twelfth International Conference on Learning Representations*

---

> ### Author Response · Authors · 2025-08-03
>
> If any of our responses above remain unclear or do not fully resolve your misunderstandings, we would be truly grateful if you could let us know as soon as possible. This would allow us sufficient time to further discuss or improve our work. Your feedback is extremely valuable to us, and we deeply appreciate your guidance and support. Thank you so much for your time and patience.

---

> ### Author Response · Authors · 2025-08-05
>
> Thank you for your time and for recognizing the value of the additional experiments. We will surely incorporate the new results into the final version.
>
> We also respect that views on novelty can differ based on individual backgrounds and tastes. Our goal is simply to present our contributions: a) the unified reasoning-ranking architecture, b) label-free RL training (RecPO), and c) the stable fused reward, as clearly as possible and to learn from the feedback of the reviewers and ACs, minimizing misunderstandings as less as possible.
>
> Besides novelty, may we confirm whether the other issues you raised, i.e., latency, baseline coverage, benchmarking result, and the benefit of reasoning—are now resolved in your view? Any final pointers would be greatly appreciated; with the rebuttal period now extended, we remain open to further explanations and discussions.
>
> Thank you again for the constructive discussion.

---

### Official Review · Reviewer_j6Fe · 2025-07-15

**Clarity:** 3
**Significance:** 2
**Originality:** 2
**Rating:** 4
**Confidence:** 4

**Summary:**

In this paper, the authors proposed R2ec, a large‑scale recommender that lets an LLM “think” and rank in the same forward pass, adding a recommendation head to the usual language‑model head so that reasoning tokens are generated before the final item prediction. It is trained with RecPO, a single‑step reinforcement‑learning scheme that combines discrete ranking rewards with a soft similarity signal, eliminating the need for costly human rationale labels.

**Questions:**

1. Have you run preliminary experiments on more dissimilar verticals (e‑g., Fashion or Electronics) or on non‑Amazon corpora to gauge cross‑domain robustness?
2. Can you quantify how often the thinking tokens contain multi‑step inference (e.g., preference trade‑offs or counterfactual statements) versus simple profile paraphrases?
3. (If possible) I'd like to see larger size LLMs

**Ethical Concerns:**

["NO or VERY MINOR ethics concerns only"]

**Final Justification:**

Thank you for the comprehensive experiments, many my concerns have been resolved.

**Limitations:**

yes

**Paper Formatting Concerns:**

NAN

**Quality:**

2

**Strengths And Weaknesses:**

Pros:
1. The topic is both intriguing and holds potential for significant developments.
2. The paper is well-organized and reader-friendly, making it easy to understand.
4. The experiment results seem promising.

Cons:
1. The datasets presented in the paper is not diverse enough. The study benchmarks R2ec on only three closely related Amazon verticals—Games, CDs & Vinyl, and Musical Instruments.
2. The so‑called thinking tokens largely paraphrase the user’s historical preferences into a short profile summary. They seldom exhibit multi‑step inference, counterfactual analysis, or explicit preference trade‑offs, so the generated text reflects descriptive recall more than genuine reasoning.
3. The authors embed the recommendation head directly in the decoder and argue this tight coupling is essential. Yet a plausible alternative—attaching an external ranking layer that consumes the hidden state after the profile summary and is trained end‑to‑end via back‑propagation—is neither implemented nor compared.

---

> ### Author Rebuttal · Authors · 2025-07-31
>
> We sincerely thank the reviewer for recognizing the intriguing nature of our study, the clear organization of the manuscript, and the promising experimental results. We also appreciate the constructive feedback regarding dataset diversity, the depth and nature of reasoning tokens, and the comparison to alternative ranking head architectures.
>
> We address the above suggestions one by one as follows.
>
> ---
>
> > Q1: Dataset Diversity
> >
>
> Considering your suggestion on “more dissimilar verticals (e‑g., Fashion or Electronics)” and the limited rebuttal time, we have added the Electronics dataset and compared R2ec with the 2 advanced baselines, namely SASRec and. D³. The performance results are summarized as follows:
>
> |  | N@5 | N@10 | H@5 | H@10 |
> | --- | --- | --- | --- | --- |
> | R²ec | 0.020483 | 0.025209 | 0.03137 | 0.046221 |
> | w/o reasoning | 0.019277 | 0.024349 | 0.029368 | 0.045053 |
> | SASRec | 0.011274 | 0.016042 | 0.016686 | 0.031537 |
> | D³ | 0.013726 | 0.01661 | 0.021358 | 0.303687 |
> | Llara | 0.018300 | 0.02123 | 0.027865 | 0.0368763 |
>
> The results show that R²ecstill performs better on the new dataset, illustrating the cross-domain robustness on diverse domains.
>
> ---
>
> > Q2: Reasoning Behavior Analysis — How Does R²ec Reasoning to Recommend?
> >
>
> To better address your insightful question regarding the reasoning behavior, we conducted an in-depth both qualitative and quantitative analysis of R²ec's reasoning patterns.
>
> We first engaged in extensive discussions with colleagues to collaboratively analyze and summarize the diverse reasoning behaviors exhibited by R²ec. Through this process, we identified 7 typical reasoning strategies. Building on this framework, we then randomly sampled 100 reasoning outputs from each test set. These samples were annotated and analyzed through a combination of detailed human evaluation and automated assessment with GPT-4.1, ensuring both depth and consistency in our analysis.
>
> 1. **Qualitative Analysis.** We found that there are 7 different reasoning behaviors for recommendation, including:
>     - Attribute Abstraction: Identifying underlying attributes of purchased items.
>     - Pattern Recognition & Clustering: Detecting frequent item patterns to form user-interest clusters.
>     - Scenario/Role-Based Reasoning: Inferring user roles or specific use-cases.
>     - Temporal & Sequential Reasoning: Using purchase timing and order for context.
>     - Self-Explanation: Providing explicit rationales for recommendations.
>     - Negative Preference Exclusion: Avoiding previously negatively-rated item types.
>     - Multi-Objective Balancing: Managing trade-offs between conflicting user objectives.
>
>     These distinct reasoning strategies help explain why and how reasoning aids recommendation. Large recommender model equipped with reasoning can better distill salient user preferences and intentions from noisy or complex behavioral data, abstract patterns beyond surface-level co-occurrence, and generate more contextually appropriate and user-aligned recommendations. This richer interpretive capacity enables the model to move beyond shallow heuristics, providing both higher accuracy and more interpretable results.
>
> 2. **Quantitative Analysis.** To further substantiate our findings, we quantified the distribution of reasoning behaviors across different datasets. As noted, each reasoning output may exhibit multiple behaviors simultaneously. The table below reports the **proportion** of sampled reasoning outputs within each dataset that display each behavior:
>
>
>     | Dataset | Attribute Abstraction | Pattern Recognition & Clustering | Scenario/Role-Based Reasoning | Temporal & Sequential Reasoning | Self-Explanation | Negative Preference Exclusion | Multi-Objective Balancing |
>     | --- | --- | --- | --- | --- | --- | --- | --- |
>     | Instruments | 0.95 | 0.99 | 0.98 | 0.22 | 1.00 | 0.07 | 0.72 |
>     | Games | 1.00 | 0.98 | 0.92 | 0.08 | 0.91 | 0.02 | 0.51 |
>     | CDs | 0.99 | 0.94 | 0.74 | 0.16 | 0.82 | 0.06 | 0.23 |
>     | Electronics | 0.97 | 0.84 | 0.94 | 0.12 | 0.97 | 0.15 | 0.76 |
>
> **Note:** Each value reflects the proportion of sampled reasoning outputs within the corresponding dataset that demonstrate a given reasoning behavior.
>
> This quantitative results highlight several key trends. First, **the distribution of reasoning behaviors differs across datasets**, indicating that after training, the model self-adapts its reasoning patterns to fit the characteristics of each domain and its user-item interactions.
>
> Second, the variability in these proportions suggests that even within a single domain, R²ec flexibly selects different reasoning strategies based on the specific user sequence.
>
> Overall, these results show that the model does not follow a fixed template but instead employs context-aware reasoning tailored to both the dataset and the individual user, contributing to improved recommendation performance and interpretability.
>
> ---
>
> > Q3: Comparison with External Ranking Layer Baseline
> >
>
> We appreciate the suggestion to compare against an alternative approach—attaching an external ranking layer to the hidden state after the profile summary and training end-to-end via back-propagation.
>
> To this end, we implemented and evaluated this design under two variants: (1) a ranking layer without explicit reasoning, and (2) a ranking layer with RecPO. As shown in the table below, both approaches result in substantially lower performance compared to our proposed tightly-coupled architecture:
>
> |  | N@5 | N@10 | H@5 | H@10 |
> | --- | --- | --- | --- | --- |
> | ranking layer w/o reasoning | 0.0017 | 0.0019 | 0.0022 | 0.0037 |
> | ranking layer with RecPO | 0.0025 | 0.0027 | 0.0030 | 0.0045 |
>
> These results underscore that a naively attached ranking layer—even with end-to-end training—fails to fully leverage the model’s reasoning capacity. In contrast, our unified design which utilizes item embedding enables deeper alignment between reasoning and recommendation, resulting in significantly better performance.
>
> ---
>
> > Q4: Larger-size LLMs
> >
>
> Thanks for your suggestion. We have conducted additional experiments with a larger backbone  Gemma-9B-it, as you recommended. The results on our benchmark datasets are summarized as follows:
>
> | Gemma-9B-it | N@5 | N@10 | H@5 | H@10 |
> | --- | --- | --- | --- | --- |
> | w/o reasoning | 0.037017 | 0.045978 | 0.059524 | 0.089286 |
> | R²ec | 0.045061 | 0.052692 | 0.063988 | 0.087798 |
>
> For comparison, we also present the results under Gemma-2B-it backbone from the original paper:
>
> | Gemma-2B-it | N@5 | N@10 | H@5 | H@10 |
> | --- | --- | --- | --- | --- |
> | w/o reasoning | 0.0321 | 0.0393 | 0.0469 | 0.0692 |
> | R²ec | 0.0388 | 0.0457 | 0.0588  | 0.080 |
>
> The results clearly demonstrate the consistent effectiveness of reasoning-augmented recommendation across model scales. With Gemma-9B-it, R²ec achieves a 21.7% improvement in N@5 compared to the version without reasoning, validating that the reasoning mechanism provides fundamental advantages regardless of parameter count. Notably, we observe that as model size increases from 2B to 9B parameters, the absolute performance gains from reasoning also increase, suggesting that larger models are more adept at leveraging reasoning capabilities to enhance recommendation quality.
>
> ---
>
> > Summary
> >
>
> In response to your suggestions, we have expanded our evaluation with additional datasets, conducted experiments with an external ranking layer, tested larger model architectures, and performed detailed analysis of reasoning behaviors in recommendation. These additions have strengthened rather than challenged our work, validating R²ec's robustness across diverse scenarios while more clearly demonstrating the value of reasoning-enhanced recommendation systems.

---

> ### Author Response · Authors · 2025-08-03
>
> We sincerely hope our responses have addressed your concerns. If there are still any issues or if further clarification is needed, please do let us know at your earliest convenience, so we may have enough time to discuss and make improvements. Your feedback is very valuable to us, and we truly appreciate your guidance and support. Thank you so much for your time and patience.

---

> > ### Comment · Reviewer_j6Fe · 2025-08-04
> > **response to the rebuttal**
> >
> > Thank you for the comprehensive experiments, many my concerns have been resolved, I'll increase the score.

---

> > > ### Author Response · Authors · 2025-08-04
> > >
> > > Thank you for all the insightful feedbacks and raising score. We greatly appreciate your time and effort, please feel free to contact us with any further questions or suggestions.

---

### Note · Authors · 2025-08-12

Dear AC and Reviewers,

We sincerely thank the reviewers for the time and constructive feedback throughout the review process. We were fortunate to have in-depth discussions with all four reviewers, through which we addressed concerns, responded questions, clarified misunderstandings, and reinforced our novelty and motivation with systematic analysis and literature references. We would like to briefly restate our work and the improvements made during rebuttal:

1. **Novelty.** Our work makes three main contributions:
    1. We develop a reasoning RL framework for recommendation that requires no reasoning annotations, representing a “DeepSeek-R1-Zero” moment for the domain, which is important because reasoning annotations from individuals are difficult to collect in recommendation.
    2. We propose a unified reasoning-based recommendation architecture that achieves lower latency than both reasoning-based and non-reasoning LLM recommender baselines.
    3. We introduce a novel fused reward mechanism (combining discrete ranking metrics and continuous similarities), as standard RL rewards from general LLM domains did not work in recommendation domain.
2. **Efficiency.** Following reviewer jU2r’s suggestion, we expanded the latency comparison to include both **reasoning and non-reasoning LLM-based recommendation baselines** under the same evaluation setting. Results show that our approach achieves **lower latency than all the above baselines**, thanks to our unified architecture utilizing dense item embeddings that eliminate large codebook decoding overhead.
3. **Reasoning Behaviors.** We conducted detailed qualitative and quantitative analyses of R²ec’s reasoning outputs, identifying seven distinct strategies (e.g., attribute abstraction, multi-objective balancing). This demonstrated how reasoning advances LLM-based recommenders with richer user preferences and item semantics beyond shallow co-occurrence patterns.
4. **Other Suggested Experiments.** We expanded our experiments to include additional backbones, datasets, baselines, and detailed ablations examining architectural and training choices. These additions make the empirical validation broader and more robust.

We are grateful for all the reviewers' thoughtful feedback, which did not alter our core motivation or approach but instead enhanced the robustness of our research and establish a more solid empirical foundation.

---

### Decision · Program_Chairs · 2025-09-17

**Decision:**

Accept (poster)

**Comment:**

The authors address the novel and interesting question of how LLMs can leverage reasoning for enhanced recommendation performance.  Reviewers agree that the paper is well written and the research question is interesting.  They raise concerns regarding potential alternative architectures, the need for more baselines (improved during rebuttal), limitations of the three related datasets (one dissimilar dataset added on rebuttal), and most importantly the need for reasoning behavior analysis.  This last question was addressed quite comprehensively and insightfully during the rebuttal phase and represents a significant novel contribution to the nascent "reasoning for recommendation" that should ideally be incorporated as a centerpiece of the main paper's empirical analysis on revision.  In light of the extensive author discussion and experimentation to address reviewer concerns, reviewers are generally in favor of acceptance post-rebuttal.  The AC concurs with acceptance with the expectation that the authors will do their best to incorporate all rebuttal discussion/content into the final version of their paper.